# Magnetic steering continuum robot for transluminal procedures with programmable shape and functionalities

Liyang Mao [1,2], Peng Yang[1,2], Chenyao Tian [1,2], Xingjian Shen[1], Feihao Wang[1], Hao Zhang [1] ✉, Xianghe Meng [1] ✉ & Hui Xie [1] ✉

Millimeter-scale soft continuum robots offer safety and adaptability in transluminal procedures due to their passive compliance, but this feature necessitates interactions with surrounding lumina, leading to potential medical risks and restricted mobility. Here, we introduce a millimeter-scale continuum robot, enabling apical extension while maintaining structural stability. Utilizing phase transition components, the robot executes cycles of tip-based elongation, steered accurately through programmable magnetic fields. Each motion cycle features a solid-like backbone for stability, and a liquid-like component for advancement, thereby enabling autonomous shaping without reliance on environmental interactions. Together with clinical imaging technologies, we demonstrate the capability of navigating through tortuous and fragile lumina to transport microsurgical tools. Once it reaches larger anatomical spaces such as stomach, it can morph into functional 3D structures that serve as surgical tools or sensing units, overcoming the constraints of initially narrow pathways. By leveraging this design paradigm, we anticipate enhanced safety, multi-functionality, and cooperative capabilities among millimeter-scale continuum robots, opening new avenues for transluminal robotic surgery.

The lumina of the digestive, respiratory, and urogenital organs or vessels of the body transport air, blood, fluids, food, and other substances inside the body or between the body and the exterior, potentially allowing invasive access to various target tissues. Using lumina's patency, transluminal procedures have been proposed, reshaping medical treatment through less invasive operations[1,2]. Small-scale medical devices[3,4] can navigate through the lumen of an organ or a vessel to surgical sites for the diagnosis and treatment of prominent abnormalities and diseases, reducing postoperative pain and the risk of incisional complications. Particularly, characterized by safety and adaptability due to their passive compliance, millimeter-scale soft continuum robots offer a promising approach to less traumatic and more flexible transluminal procedures, thanks to innovative developments in microfabrication techniques[5,6],

propulsion strategies[7,8], distal actuation methods[9–11], and mechanics-based kinematic models[12,13].

However, even the state-of-the-art small-scale soft continuum robots with actively steering tips necessitate interactions with surrounding lumina. Their bodies can only be passively flexed, relying on the force generated by interactions with the luminal walls[14,15]. This results in a range of challenges, from potential medical risks to restricted mobility. One pressing concern is the potential damage to fragile tissues in tortuous lumina due to the robot's interactions with luminal walls, especially in complex, sharply curved pathways[16,17]. Another issue is the accumulated frictional resistance created by the forces described above, especially in tortuous vessels, which could inhibit movement and result in the sudden catastrophic release of

[1]State Key Laboratory of Robotics and Systems, Harbin Institute of Technology, 2 Yikuang, Harbin 150001, China. [2]These authors contributed equally: Liyang Mao, Peng Yang, Chenyao Tian. ✉e-mail: haoz@hit.edu.cn; mengxianghe@hit.edu.cn; xiehui@hit.edu.cn

stored elastic energy[9]. The thermal response variable stiffness magnetic continuum robot has the advantages of small scale, flexibility, and accessibility[18–21]. It can reduce stiffness during deployment to minimize forces on tissues, but the forces still accumulate as the robot continues to deepen the lumen. This not only results in the risks described above but also leads to localized buckling, preventing the transfer of thrust from proximal to distal. Moreover, A further challenge arises when the robot navigates into open areas such as the heart, stomach, bladder, or vascular junctions, where its mobility becomes severely restricted due to the absence of anatomical structures for interaction[22,23].

Therefore, continuum robots should have the ability to actively comply with the features of the lumina and reach the target independently of tissue interaction. Robots exhibiting the 'follow-the-leader' (FTL) behavior offer a promising solution, as they can operate without relying on environmental interactions. This behavior enables FTL continuum robots to navigate through the patient's anatomy adeptly while avoiding delicate regions. FTL behavior, as herein defined, denotes the movement of a robot body along a trajectory guided by its tip[24], which can be realized by integrating multiple independent active deformation segments to enhance degrees of freedom (DOF). However, the challenge lies in arranging these deformation segments as efficiently as biological muscles, consequently limiting FTL deployment to specific or pre-defined trajectories[25–27]. An alternative approach to FTL continuum robotics involves the deployment of two concentrically arranged snake robots with a shape-lock mechanism[28]. Nonetheless, the intricate locking structure results in increased cross-sectional dimensions and restricted angulation range.

Here, we put forth a millimeter-scale continuum robot with FTL behavior, capable of apical extension and secure structural stability (Fig. 1a, b). Utilizing a dual-component system based on phase transitions, the robot engages in periodic, tip-based elongation steered by a programmable magnetic field. Each of these motion cycles integrates a stable, solid-like backbone along with a liquid-like component for forward advancement, allowing the robot's shape to be actively programmed or reprogrammed through trajectory planning of its tip, independent of environmental interactions. When integrated with advanced imaging technologies, our robot is capable of achieving precise magnetically guided navigation, akin to a bine threading through narrow and intricate lumina, while significantly reducing tissue damage and friction (Fig. 1c i). Its mobility characteristics, including accessibility and dexterity, are unrestricted in open spaces (Fig. 1c ii, iii). In addition to executing surgical functions through the incorporation of microsurgical tools (Fig. 1c iv, v), our robot is not merely a tool-carrier. Upon reaching expansive anatomical areas like the stomach, it has the capability to morph in situ into complex 3D structures that serve as either surgical instruments or sensing units. This overcomes the limitations of the natural narrow lumina in surgical tools' geometries and functionalities. Uniquely, the robot can also sense external factors like pressure and self-form knots to enhance its sensitivity (Fig. 1c vi). We substantiate these capabilities through both ex vivo and in vivo studies, demonstrating the robot's mobility, functionalities, and compatibility with existing medical technologies. Table 1 compares the performance of our robot with existing continuum robots including commercial endoscopes.

## Results

### Working principle

The detailed structure of our continuum robot is depicted in Fig. 1a. We achieved the FTL behavior of the proposed continuum robot by alternately advancing a couple of phase transition components (PTCs) that are constrained to move only axially relative to each other (Fig. 1b). These PTCs can transition between solid and liquid states, altering their stiffness to switch roles between support and movement. The PTC capable of active steering and navigation under the applied

magnetic field is named the Guider, and the other following the trajectory of the Guider is called the Follower. Low melting point alloy (LMPA) was selected as the phase transition material, and magnetic actuation was used to navigate the robot. LMPA and heating circuit (Fig. S1a) were encapsulated by silicone tubes to obtain the primary PTC[18,29], which can be fabricated with diameters of a few millimeters and lengths of several meters (Fig. S1b). The internal pressure adds self-healing properties to the PTC (Fig. S1c). Figure 2a illustrates the preparation process of the robot (see detailed preparation process in Materials and Methods). The Guider can be obtained by embedding a tiny permanent magnet in the tip of the PTC, and the Follower can be obtained by bonding a silicone tube along the axial direction of the PTC (Fig. 2a v). The Guider was mounted coaxially with the silicone tube of the Follower, allowing the two PTCs to slide only axially against each other. The friction that prevents relative sliding can be significantly reduced by curing hydrogel on the surface of the Guider and the Follower (Fig. 2b).

The Guider and Follower operate in an alternating manner, forming a cooperative relationship as follows: (1) Initially, the Guider, heated to the phase-transition temperature, becomes flexible for forward movement, while the Follower remains rigid to provide support. With multiple orders of magnitude differences between their flexural strengths (Fig. 2c, detailed structure analysis provided in Materials and Methods), the rigid Follower maintains the shape of the deployed segment of the continuum robot, serving as a conduit to deliver the flexible Guider forward under proximal thrust. The flexible outstretched segment of the Guider will be deflected toward the direction of the magnetic field. (2) Subsequently, the Guider and Follower switch states, transitioning their roles within the motion cycle of the robot. The now rigid Guider stabilizes the deployed shape, while its protruding segment forms a new conduit, constraining the shape of the propelled flexible Follower. The maximum cross-sectional size of the robot is ~3–4 mm, with a potential length extending up to meters. For demonstration purposes, we employ a robot with a length of ~400 mm (Fig. S1d).

### Thermal characterization and management

In the process of thermal management optimization, to obtain the minimum current required to heat the LMPA to the phase transition temperature in a fluid environment (water and air) at 37 °C, we simulated the stabilization temperature and the time to reach the melting point as a function of the magnitude of the current flowing through the resistive heater (Fig. S2a). The results show that due to the high thermal conductivity of water, a current of at least 0.9 A is required to change the stiffness of PTCs in water, while only 0.3 A is required in air (Fig. 2d). LMPA requires a response time of a few seconds in both environments. Although the increase in current can lead directly to a reduction in response time, the compact distance between the two PTCs causes the Joule heat to rapidly soften the other rigid PTC as well through heat transfer, resulting in the collapse of the entire shape. We simulated the correspondence between the average temperatures of the Guider and Follower when they were heated separately in a fluid environment at 37 °C (Fig. S2b) and obtained the maximum threshold of the two PTCs' temperatures to keep the entire structure (Fig. 2e and Fig. S2c).

Proportional-integral-derivative (PID) control was applied to accurately control PTC temperature based on the mapping of temperature $T$ to PTC total resistance change $\Delta R_{tot}$ (the change in total resistance relative to 37 °C)[18]. A sourcemeter was used to heat the circuit and detect the resistance. To investigate the relationship between the temperature and the change in total resistance, the PTC, without electrical heating, was placed in a thermostatic water bath, where the temperature of the PTC could be accurately set by changing the temperature of the water. The total resistance of the PTC corresponding to each temperature was captured by the source meter,

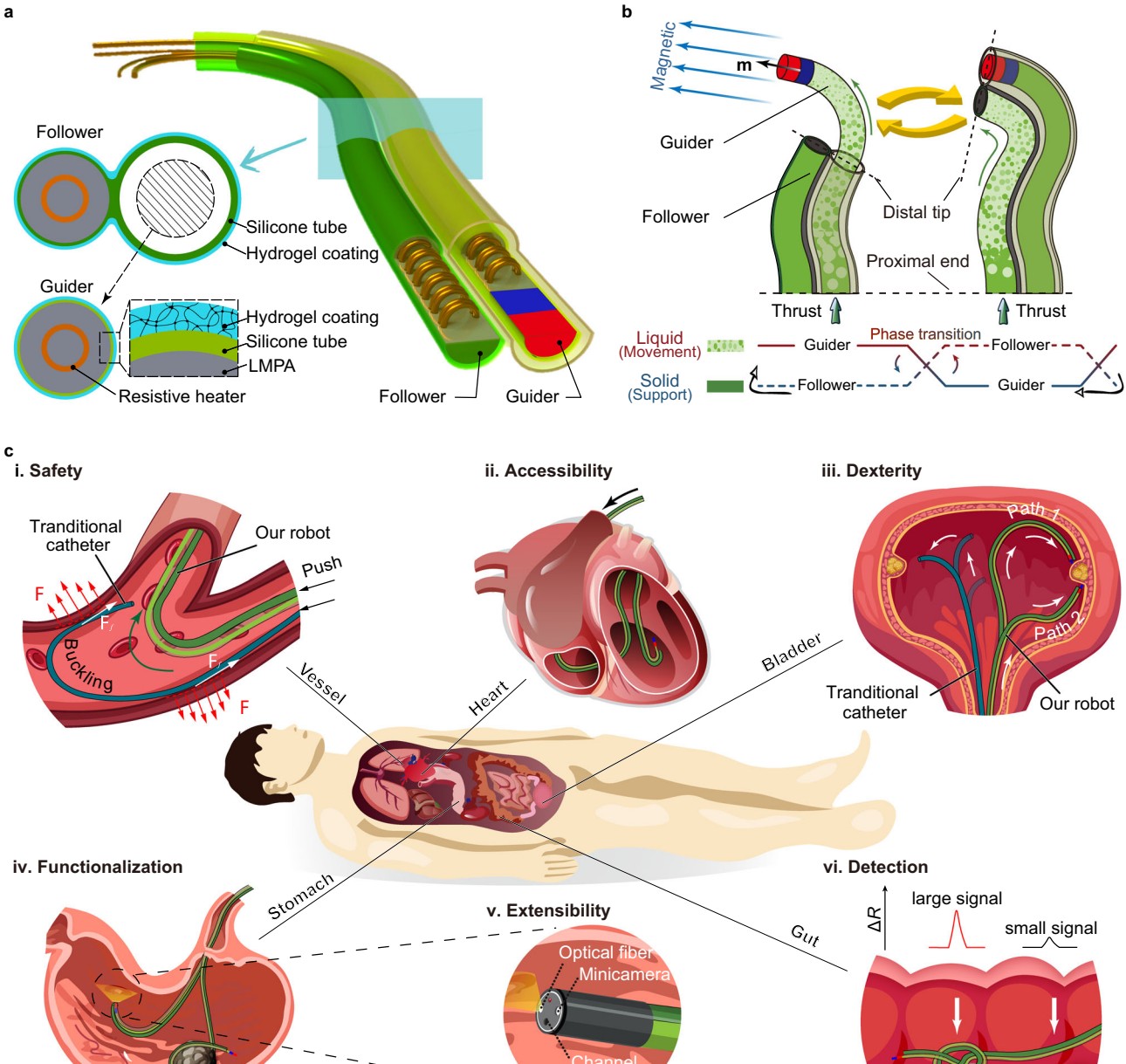

**Fig. 1 | Schematic of the millimeter-scale magnetic steering continuum robot for transluminal procedures. a** Overview of the proposed continuum robot. Our robot consists of a couple of phase transition components (PTCs) named "Guider" and "Follower". The Guider is assembled coaxially in a silicone tube bonded to the Follower, and its tip has an embedded tiny permanent magnet to respond to external magnetic fields. The low melting point alloy (LMPA) and the resistive heater are encapsulated by the silicone tube, and the hydrogel coating is grown on the surfaces of the Guider and Follower to reduce the friction coefficient significantly. **b** Schematic of our robot's design principle. The shapes of the Guider and Follower are interlocked. In the motion cycle, the two PTCs alternate between solid and liquid, corresponding to the states of support and movement. **c** Under magnetic steering, the magnetic steering continuum robot shows excellent movement performance (**i, ii, iii**) and functionalities (**iv, v, vi**) in lumina or the relatively open cavities of organs.

subtracting the resistance in 37 °C to obtain Fig. 2f. We considered only the completely rigid or flexible PTC case and adopted 4.3 mΩ and 31.8 mΩ as the bounds of complete phase transition. To record the $\Delta R_{tot}$ over time (Fig. 2g), a PTC was electrically heated to be completely flexible and maintained at that temperature under PID control in a 37 °C environment (37 °C water bath or thermostat) and then naturally cooled to rigidity. The PTC can transform from a rigid into an utterly flexible state in ~5 s in air and 10 s in water. Stiffening ends about 15 s and 10 s after the start of cooling in air and water, respectively (Fig. S3 d, e and Supplementary Movie 1). As can be seen from the heat transfer simulation (Fig. S3 a, b, c), theoretically, a completely flexible PTC can cool down to a completely rigid state quickly. The difference between

the experiment and the simulation may lie in the deviation of the robot preparation as well as the difference in the experimental environment. The surface temperature of the flexible PTC can be stabilized at 42 °C (Fig. S2d) within a biocompatible temperature range (i.e., below 50 °C)[30]. Moreover, axial deformation does not affect the total resistance, which means that the bending of the PTC due to magnetic torque does not disturb the relationship between temperature and total resistance change.

## Periodic numerical control of the robot for deployment

Experimental equipment includes the magnetic actuation system, the advancement unit, and the sourcemeter (Fig. 3a). The Guider is first

## Table 1 | Performance comparison of our robots with existing continuous robots

| | Ref. | Actuation strategies | Diameter (mm) | Length (mm) | Angulation range (°) | Variable stiffness | FTL deployment | Functionality | Application Scope | In vivo validation |
|---|---|---|---|---|---|---|---|---|---|---|
| **Variable Stiffness Robot** | Kim[10] | Magnetic | 0.8 | ~200 | ±161 | NO | NO | Carrying tools | Vascular intervention | **Yes** |
| | Lussi[18] | Magnetic | 1 | 30 | ~±180 | Thermal response | NO | Carrying tools | Ophthalmic Surgery | NO |
| | Mattmann[19] | Magnetic | 1.3 | 140 | - | Thermal response | NO | Carrying tools | Stomach and vessels | NO |
| | Chautems[20] | Magnetic | 2.33 | 50 | ~±180 | Thermal response | NO | Carrying tools | Vascular intervention | NO |
| | Piskarev[21] | Magnetic | 2 | 88 | ±51 | Thermal response | NO | Carrying tools | Vascular intervention | NO |
| **FTL Robot** | Gao[26] | Tendon-driven | 3.4 | 60 | ~±180 | NO | Limited path | - | Brain surgery | NO |
| | Gilbert[25] | Concentric tube | 2.18 | 120 | - | NO | Limited and pre-defined path | - | Brain surgery | NO |
| | Peyron[27] | Concentric tube + Mag | 1.07 | 213.5 | ±136.4 | NO | Limited and pre-defined path | | Nasal cavities and ear canals | NO |
| | Kang[28] | Multi-backbone | 30 | <200 | ±90 | Mechanical locking | **Arbitrary path** | - | - | NO |
| | Amanov[47] | Tendon-driven | 7 | 165 | - | NO | Limited path | - | - | NO |
| **This work** | | Magnetic | <4 | >400 | ~±180 | Thermal response | **Arbitrary path** | **Formation of functional structure** or carrying tools | Vascular intervention and gastro-intestinal surgery | **Yes** |
| **Commercial Catheters** | Video cholangioscope (Innovex Medical Co.) | | 5 | 380 | ±180 | NO | NO | Carrying tools | Gastrointestinal tract | **Yes** |
| | Video bronchoscope BF-1TH190 (Olympus Co.) | | 6.2 | 600 | ±120 | NO | NO | Carrying tools | Bronchial tube | **Yes** |
| | Fiber pharyngoscope FS2 (Orlvision Co.) | | 2.9 | 300 | ±130 | NO | NO | Carrying tools | Nasopharynx | **Yes** |
| | Video cysto-nephroscope CYF-VH (Olympus Co.) | | 5.5 | 380 | −130 ~ 220 | NO | NO | Carrying tools | Urology | **Yes** |

Categories on the left side of the table have been highlighted in bold font. Compared with existing continuum robots, our robot, with small scale and flexibility, can achieve FTL deployment along arbitrary paths. In addition to carrying tools, our robot can also form itself into functional structures in situ. The diameter, length, and angulation range of our robot have reached the level of commercial endoscopes. The advantages of our robot have been highlighted in bold within the table. Through experiments, we found that our robot exerts 100 times and 10–15 times less force on the sidewall compared to commercial catheters and conventional variable stiffness robots, respectively.

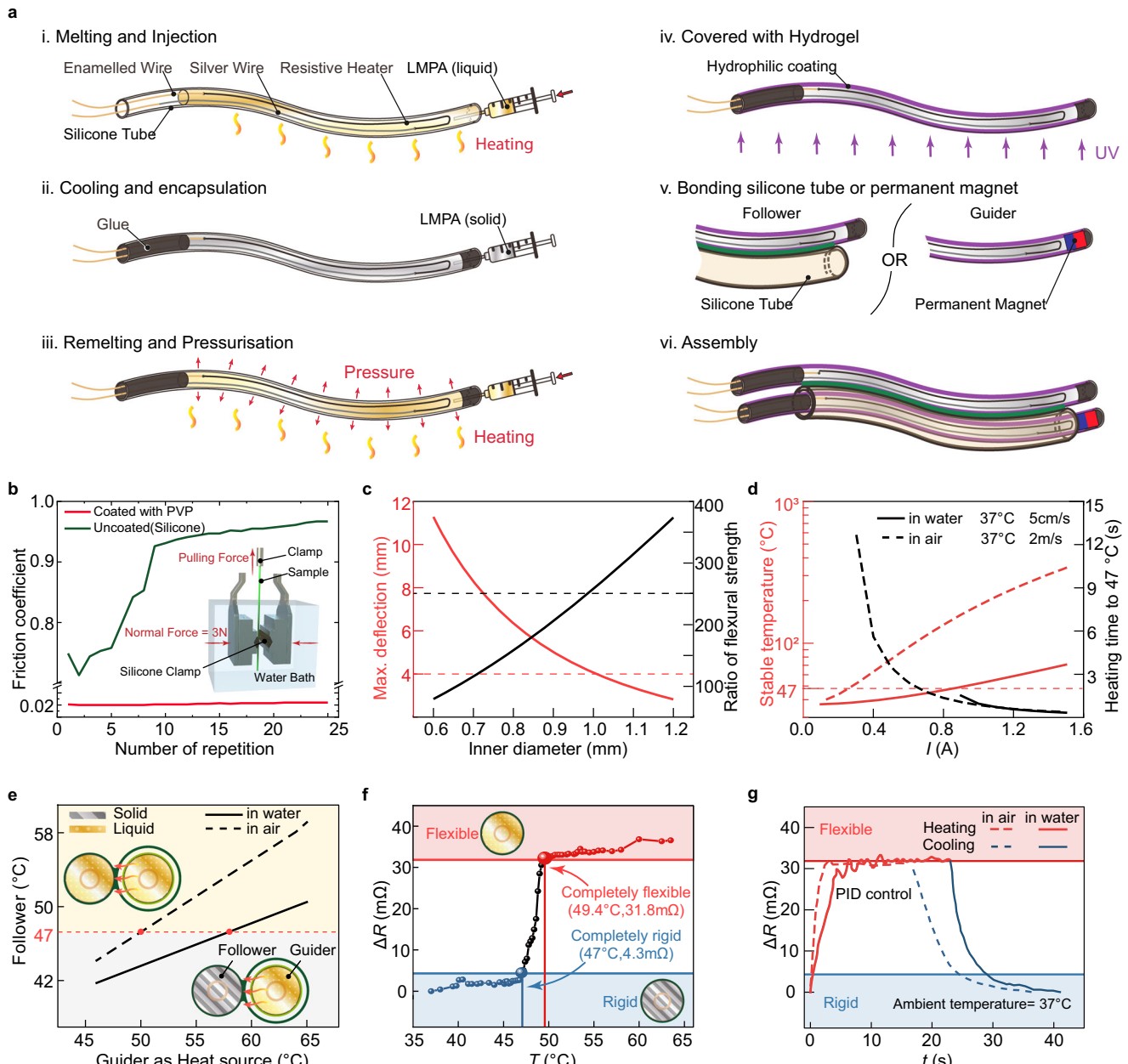

**Fig. 2 | Preparation, structure analysis, and thermal management of the robot.**
**a** Detailed preparation process. (**i**) Inject the molten LMPA into a silicone tube
where a heating circuit has been pre-arranged. (**ii**) Cool the LMPA and encapsulate
both ends of the silicone tube with glue. (**iii**) Remelt the encapsulated LMPA and
pressurize the silicone tube to maintain force on LMPA, and obtain the phase
transition component (PTC). (**iv**) Apply the hydrogel layer for lubrication. (**v**)
Mount the permanent magnet to PTC to obtain the Guider. Axially bond a silicone
tube to PTC to obtain the Follower. (**vi**) Assembly of the Guider and the Follower
results in the proposed magnetic steering continuum robot. **b** The friction coeffi-
cient is dramatically reduced by applying hydrogel to the PTC surface. The
embedded image is the schematic of the friction coefficient test. **c** In gravitational
fields, limited by the elastic modulus of the material and the application purpose,
the rigid PTC should be over 1 mm in diameter to constrain the other flexible PTC
and maintain its shape. **d** Heat transfer simulation shows that a constant heating
current maintained over 0.3 A or 0.9 A allows the LMPA to rapidly reach the phase
transition temperature in gas or liquid environments. **e** Once one PTC is heated to
flexible, the temperature of the other rigid PTC may exceed the phase transition
temperature due to heat conduction; hence the need to control the heating tem-
perature. The influence of the heated Guider's stable temperature on the Follower
was obtained by numerical simulation. **f** The mapping between temperature and
absolute resistance change is established. **g** The Guider and Follower's tempera-
tures can be proportional-integral-derivative (PID) regulated in air and water
(ambient temperature = 37 °C).

pushed out of the silicone tube on Follower by the advancement unit,
then heated and softened by the sourcemeter, and finally deflected by
the magnetic field generated by the magnetic actuation system. Under
the constant curvature assumption, the deformation of the Guider's tip
under a magnetic field can be quickly calculated (see kinematic mod-
eling in Materials and Methods for details). To minimize the effect of
gravity, the Guider should not advance more than 40 mm at a time

(Fig. 3b). To maintain movement efficiency, the advancing distance in
each step should be no less than 20 mm. Consequently, the accessible
area and inclination angle of the Guider's tip in a motion cycle can be
obtained (Fig. 3c).

Since the entire shape of the robot follows its tip trajectory, once
the length of advance *l* and the magnetic flux density **B** in each motion
cycle have been calculated, the robot can be deployed along the

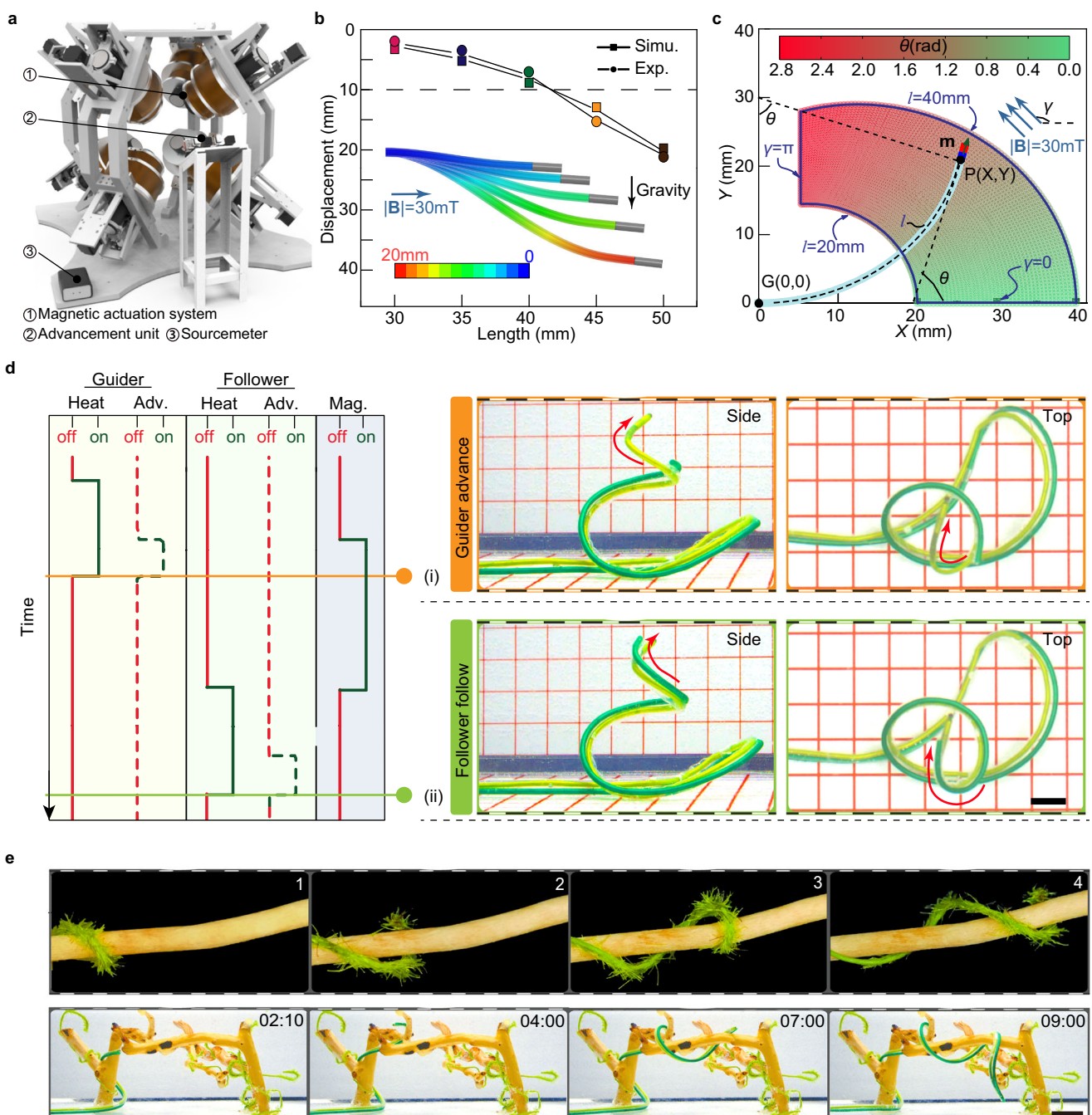

**Fig. 3 | Numerical control of the robot. a** Schematic of the experimental equipment. **b** The scenario was analyzed where the effect of gravity is most prominent: the softened Guider is placed horizontally as a cantilever beam under horizontal magnetic and vertical gravity fields. **c** Under the constant curvature assumption, the accessible area of the Guider ($l \in {}^{20,40}$ mm) is highlighted in blue at 30 mT magnetic flux density ($\gamma \in [0, \pi]$ rad). The background color represents the tip deflection angle as indicated by the color bar in rad. **d** Our robot navigated under a series of pre-planned magnetic fields and formed a predetermined body shape. Adv. and Mag. abbreviations for advance and magnetic Field. Scale bars, 10 mm. **e** Comparison of the movement of the bine and our robot. Pictures document that the continuum robot curls around the support in an aqueous environment like a natural climbing plant: bine. Scale bars, 10 mm.

planned path. The motion of the robot in one cycle is shown in Fig. 3d. The robot navigated under a series of pre-planned magnetic fields and formed a predetermined body shape (Supplementary Movie 2). The time sequence of the magnetic actuation system, the advancement unit, and the sourcemeter was planned ahead of time. In the motion cycle, the Guider first undergoes a controlled heating process to achieve flexibility under PID control. Once its total resistance change stabilizes, indicating the attainment of the flexible state, the Guider is propelled forward by the advancement unit and steered in response to the applied magnetic field. Subsequently, both the advancement unit

and the heating circuit are deactivated, while the magnetic field is maintained until the Guider transitions from flexibility to rigidity. Following this, the Follower is heated to flexibility and advanced along the trajectory of the Guider by the advancement unit. Acknowledging potential errors in the kinematic model and practical contingencies, priority is accorded to operator inputs to the experimental equipment during pre-programmed movements of the robot. Ultimately, our robot demonstrates the capability of follow-the-leader (FTL) deployment along arbitrary paths, exhibiting accessibility and dexterity reminiscent of a bine. It can even navigate complex terrains, including

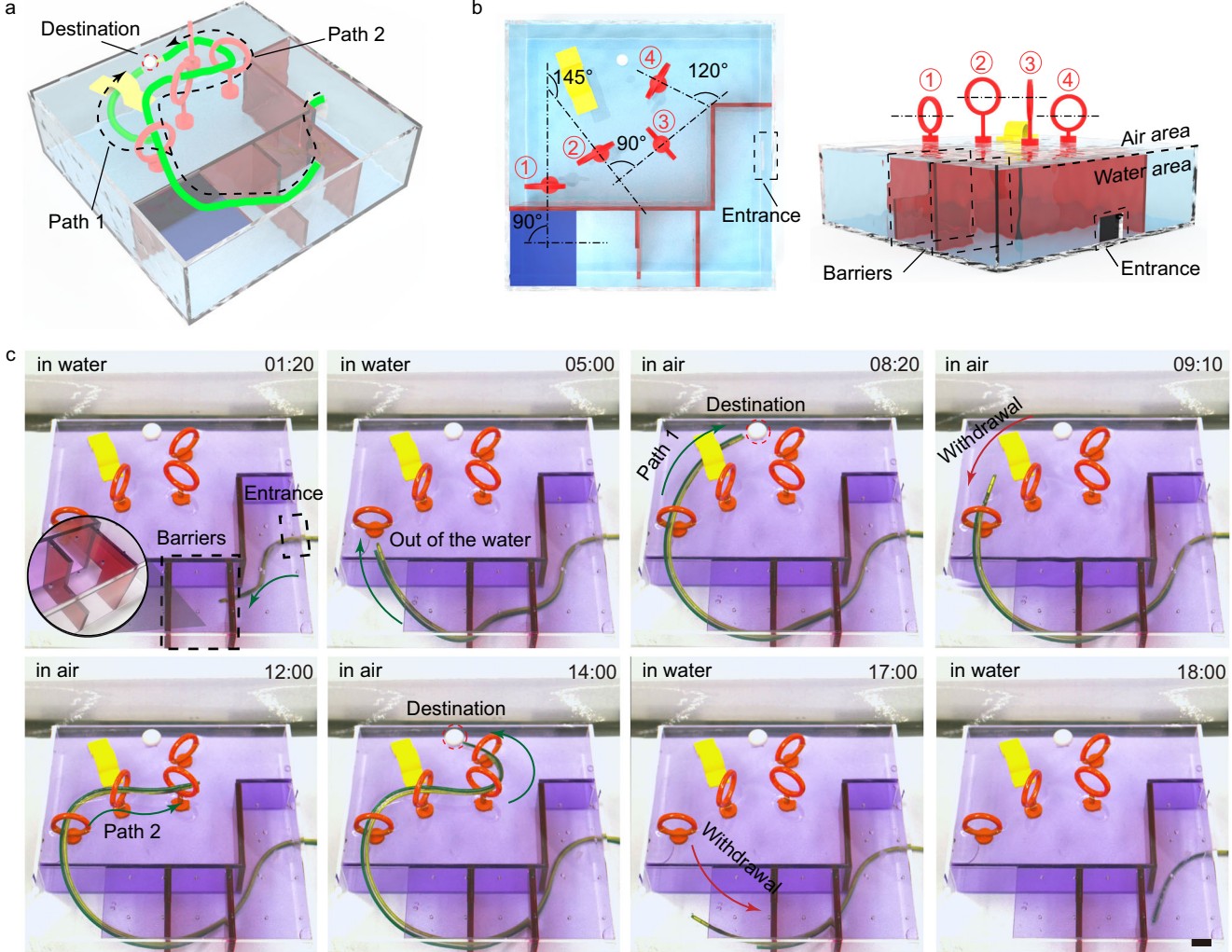

**Fig. 4 | Magnetic navigation in a complex environment. a** Schematic of the robot performing tasks through a highly unstructured environment. **b** Schematic of the working environment. The first layer with barriers is in the aquatic environment, and the second layer with a set of rings and a channel is in the air environment. There is a large angle between adjacent small rings. **c** The robot can pass through the unstructured environment without relying on interactions with it. First, the robot passes smoothly through the barriers on the first layer (0–5 min). Then it can navigate tortuous paths 1 (5–9 min) and 2 (9–18 min) to reach the region of interest. Scale bars, 10 mm.

climbing and curling around branches (Fig. 3e and Supplementary Movie 3). As can be seen in the video, although the forces between the Guider and Follower increase, due to the twisting caused by the asymmetric extension of the effective cross-section when curling, this does not hinder the relative sliding of the two PTCs, since the hydrogel considerably reduces the coefficient of friction.

**Magnetic navigation along planned paths to the destination**
Under magnetic steering, the proposed continuum robot, capable of FTL behavior, can navigate along planned paths through highly unstructured environments with little or no interactions (Fig. 4a). The working environment was divided into two layers: water and air. The robot needed to pass through the barriers in water to enter the second layer and then navigate through a channel in path 1 and a set of rings in path 2 to reach the destination twice in air, respectively (Fig. 4b). The applied magnetic field and the forward distance in each motion cycle were calculated in advance. The forward distance is adjusted to the complexity of the environment to balance the accuracy and efficiency of movement. The experimental demonstration of the fabricated prototype is shown in Fig. 4c and Supplementary Movie 4. The magnetic steering continuum robot was able to follow a planned trajectory through the constrained environment in water to the top entrance.

After entering the second layer in air, the robot followed path 1 to the destination and retracted to the top entrance. Then the robot followed path 2 through a series of loosely arranged circles to the destination again. The large angle between adjacent small rings makes it hard for existing small-scale soft continuum robots to pass through without causing pressure on the rings (Fig. 4b Left). Finally, the advancement unit withdrew the deployed robot along the deployment path. Sometimes, the robot has to make localized contact with the environment to resist gravity-induced deformation due to long cantilevers, but the weight of the millimeter-scale robot does not result in tissue damage or non-negligible frictional resistance.

**Formation of functional structures in situ**
Our focus extends beyond the inherent challenges and requirements faced by traditional soft continuum robots. We also explore the unique capability of our robot to actively and dynamically program and reprogram its entire body in situ, transforming into a functional instrument. Upon reaching the target site safely, such as the bladder, ventricles, or abdominal cavity, the robot can continue to progress along a predetermined path, morphing into complex and functional structures suitable for various surgical tasks or sensing applications. This capability allows the robot to overcome the limitations, that the

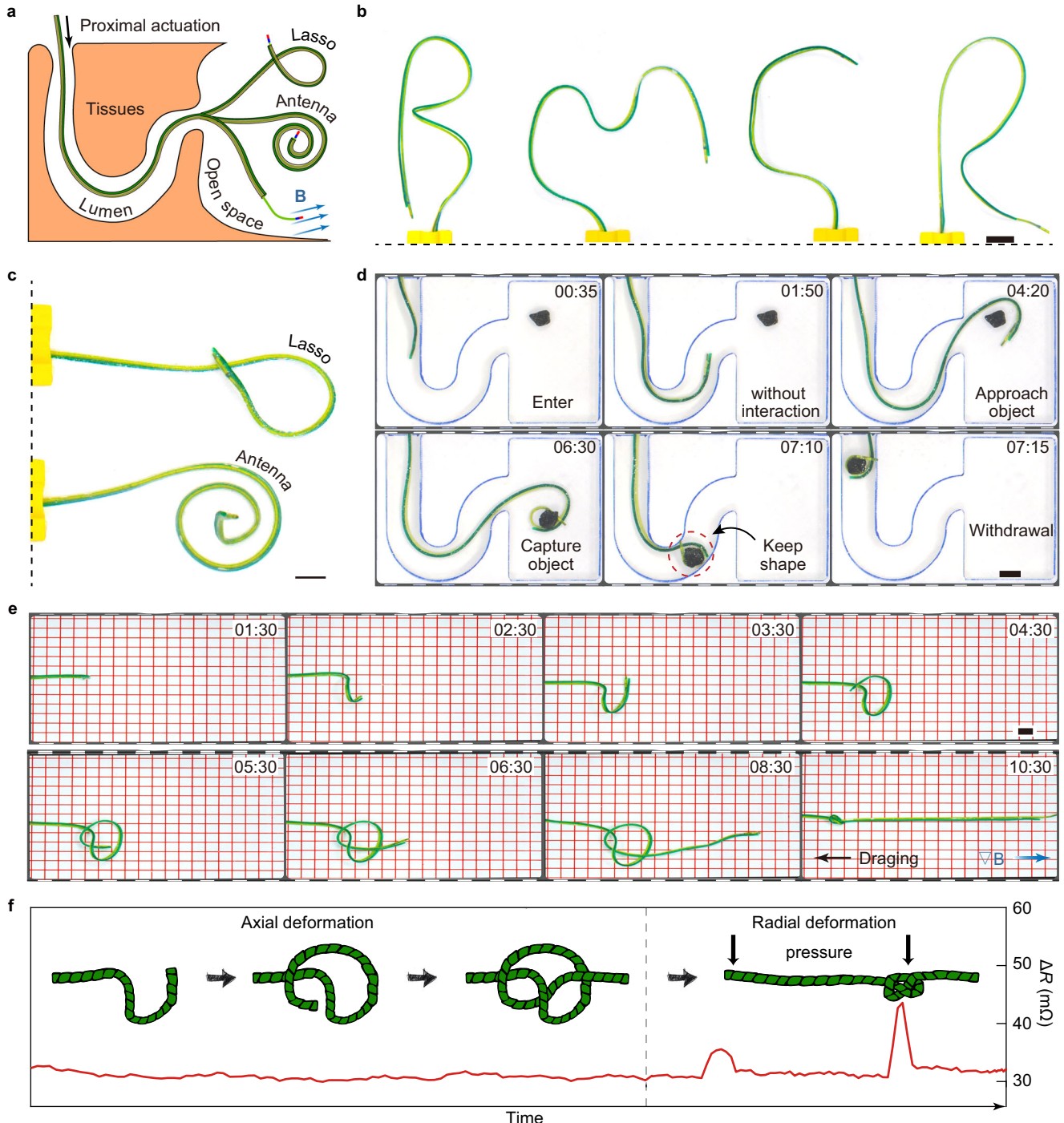

**Fig. 5 | Formation of various functional structures. a** Schematic of the robot forming large functional structures upon passing through the narrow body lumen and reaching the relatively open space. The robot can form structures such as (**b**) the letters 'B', 'M', 'C', and 'R', (**c**) a lasso, and an antenna. **d** The lasso formed by the robot, capable of segmented variable stiffness, can be used for object capture. **e** Series of optical images document that the robot can form and tie three-dimensional knot patterns tightly. All scale bars, 10 mm. **f** The robot can tie itself into a knot for pressure detection. In contrast to radial deformation, the resistance of the PTC is non-sensitive to axial deformation. The Guider and Follower only deform axially during regular operation, which causes no interference with resistance-based real-time condition monitoring. The knot improves the robot's sensitivity to radial deformation, enabling this robot to act as a pressure sensor for intestinal pressure detection.

small inner diameter of natural lumina or access ports imposed, on the geometries and functionalities of surgical tools (Fig. 5a). Compared with structures employing strategies based on elastic deformation or origami[23,31,32], our robot can form and re-form multiple functional structures in situ as needed without prior programming. Moreover, it has the rigidity to withstand disturbances to perform reliable surgical procedures that require large force and high accuracy.

Coming out of the lumen, the magnetic steering continuum robot can form various simple structures in situ as demonstrated by letters ('B', 'M', 'C','R'), a lasso, and an antenna (Fig. 5b, c, and Supplementary Movie 5). To achieve the functionality of the lasso, we added the capability of segmental variable stiffness to the robot. The internal heating circuit of the PTC is shown in Fig. S4a: an additional resistive heater was added to the circuit. By selecting the resistive heater, the

length of the variable stiffness segment of the PTC can be adjusted. Thus, the robot maintained the shape of the rigid anterior segment to capture the object when withdrawing (Fig. 5d and Supplementary Movie 6). Moreover, the initial resistive heater had a segment folded back at the tip of the PTC, which allowed segment 1 to be heated with twice the power of segment 2 (Fig. S4a). The current can be adjusted to make segment 1 flexible while segment 2 remains rigid (Fig. S4b and Supplementary Movie 7).

We also demonstrated the ability of the robot to form relatively complex functional structures (Fig. 5e and Supplementary Movie 8): the robot first moved along a planned trajectory to form a loose knotted structure, and then its entire body became flexible and was pulled taut by force generated by the advancement unit and the gradient magnetic field. We used a $50 \times 50 \times 50$ mm permanent magnet (N52) (Fig. S5a) to generate the gradient field. Magnetic flux density and magnetic flux density gradient around the permanent magnet were measured (Fig. S5b). The magnetic flux density gradient was a minimum of 1.5 T/m and a maximum of 23.1 T/m along the axis within 50 mm from the surface of the permanent magnet. We found through simulation that within 50 mm, the force could be generated to tie a knot (Fig. S5c). The knot can increase the sensitivity of the total resistance to radial deformation from pressure (Fig. 5f), which confers the potential for the robot to act in situ as a pressure transducer[33]. We measured the sensitivity of the knot to the radial force (Fig. S6a, b). The applied force and the amount of resistance change were linear within the range of 0–0.4 N ($k = 41.24$ mΩ/N, $R^2 > 0.985$) (Fig. S6c). It is, to the best of our knowledge, the only continuum robot so far that can actively tie itself in knots.

## Potential clinical applications assisted by ultrasound imaging

To validate the potential for clinical translation, we combined the robot with common medical technologies to demonstrate the navigation of our robot in biologically relevant phantoms, organs ex vivo, and environment in vivo. First, we extended the demonstrated magnetic navigation in FTL manner to a more realistic, clinically relevant environment in vitro, assisted by ultrasound imaging. To this end, we used a scaled-down resin vascular model that replicated the complex anatomy around the aortic arch, placed in a 37 °C liquid environment. The robot followed the planned path from the arteriae radialis into the common carotid artery to create a channel for neurointervention. This path had continuous large angular turns at the aortic arch that would make it difficult for conventional catheters to pass through (Fig. 6a). The digital model of the phantom vasculature was first reconstructed by multiple parallel scans of an ultrasound (US) probe clamped by a 4-DOF platform (Fig. S7). The path was then planned in the digital model, and control parameters in each motion cycle were calculated based on the kinematic model. As the robot navigated along the planned path autonomously, the ultrasound probe repeatedly scanned the same path to update the robot's position in the vessel (Fig. 6b and Supplementary Movie 9) (see Supplementary Materials). The images acquired by the camera and US indicate that the robot passed through the tortuous vessel with no interactions with the vascular wall, reducing the risk of interventional procedures. Active deformation without relying on interactions with the vessel prevents the distal end of the robot from popping forward due to the unpredictable release of built-up forward pressure accumulated in frictional contact at multiple vascular bends, making the robot suitable for interventional procedures, such as cardiac radiofrequency ablation, thrombus removal, and aneurysmal embolization.

## Potential clinical applications assisted by endoscopic imaging

We also explored the functionalities of the robot for clinical applications. This robot can actively form a hook with a size larger than the entrance size to grasp foreign objects in the gastric model (Fig. S8 and Supplementary Movie 10). The segmental variable stiffness kept the

shape of the functional unit unchanged during withdrawal. Meanwhile, we characterized the precision and accuracy of the continuum robot by testing its localization accuracy for reaching the target position several times ($n = 5$) in a row (Fig. S9a and Supplementary Movie 11). Figure S8b demonstrated that with good visual feedback, the robot's localization accuracy (Error + SD) was $0.791 \pm 0.291$ mm and the motion trajectory remained essentially unchanged over multiple deployments. As an example relevant to potential medical applications, we equipped the robot with a working channel (ID = 0.5 mm) and a minicamera with built-in illumination (OD = 1.2 mm) to demonstrate the concept of diagnosis and treatment of gastric lesions (Fig. 7a and Fig. S10a). The supplementary tools are loaded in the Guider's tip, which keeps the overall size from increasing (Fig. S10b). Figure 7b is the experimental equipment for the demonstration and the practical environment containing the porcine stomach and esophagus (see Supplementary Materials). A minicamera was pre-fixed in the stomach to provide global images of the experiment. In the clinic, another robot can be used to arrange the minicamera in a suitable stance at the desired location to provide global images (Fig. S10c), because the magnetic field does not affect the rigid robot's shape in the confined environment in vivo (Fig. S10d). Aided by on-board minicamera imaging, the robot guided by a magnetic field was able to easily slide through the esophagus into the relatively open stomach (Fig. 7c and Supplementary Movie 12). After finding the lesions (indicated by the red quadrilateral and triangle) in the stomach, the distal end was sequentially deployed close to the lesions, and the drug (indicated by blue and red ink) was sprayed onto the lesions through the manipulation channel. In practical medical applications flow control devices can be used to precisely control the amount of drug. The distal end finally moved toward the stomach antrum region to examine tissue near the pylorus. These tasks demonstrate that the distal end of the robot can be repositioned at different angles to accommodate further surgery[34,35] or minimally invasive bioprinting[36,37].

## Clinical applications in/ex vivo assisted by X-ray imaging

We further validated the potential for clinical applications of the magnetic steering continuum robot with the aid of X-ray imaging. The robot entered the porcine cadaver (50 kg; purchased from Harbin Veterinary Research Institute, China) through a vascular incision and passed through the inferior vena cava into the right atrium, which could potentially be applied in cardiac radiofrequency ablation (Fig. 8a). First, the vessels were perfused with iohexol contrast agent and imaged by X-ray. The robot was then inserted into the vessels and passed through the inferior vena cava into the thoracic cavity. Positioned by X-ray imaging, the robot can advance through the superior vena cava for neurointervention or can be deflected into the right atrium for cardiac interventions under the influence of a magnetic field generated by a permanent magnet (Fig. 8b and Supplementary Movie 13). As seen in the radiographic images, the robot easily passed through a large angular turn into the right atrium, reducing the intervention's difficulty and allowing the interventionalist to focus on subsequent sophisticated surgical maneuvers.

Using X-ray imaging, we also verified the capability of the robot to form functional units in the stomach of a live porcine model in vivo. Although Pronase Granules were swallowed preoperatively to inhibit the digestive juices secreted in the stomach, food residues still resulted in an impaired field of view. Therefore, we did not equip the minicamera for in vivo experiment but used X-ray imaging instead. As shown in Fig. 8c, the pig was anesthetized with vital signs by infusion. The pig was then transferred to X-ray imaging equipment. The operator first inserted a silicone tube coated with hydrogel into the porcine esophagus using a laryngoscope. The robot was then inserted through the silicone tube into the gastrointestinal tract and passed through the cardia into the stomach. As shown in Fig. 8d and Supplementary Movie 13, the robot was guided by the magnetic field

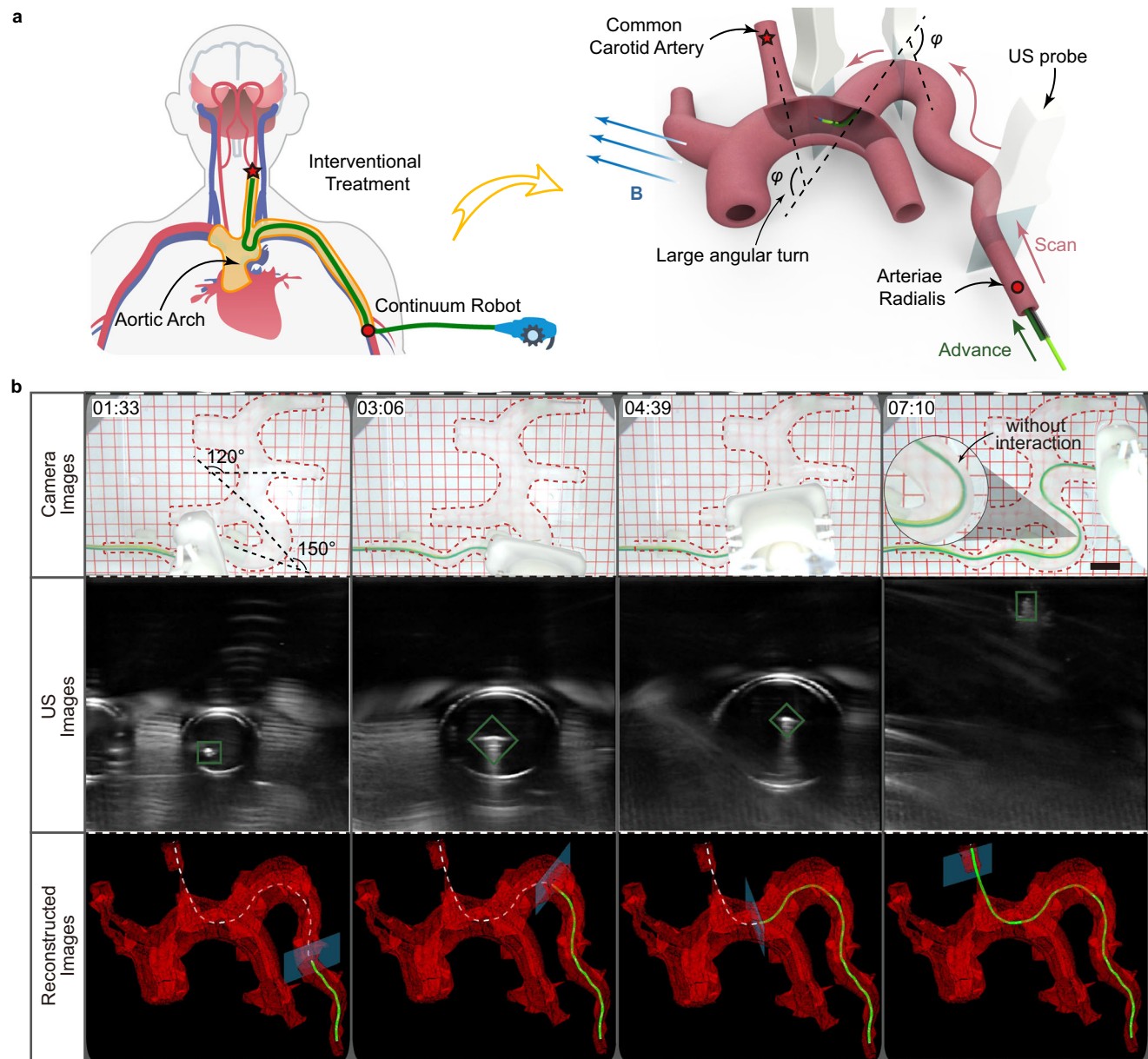

**Fig. 6 | Magnetic navigation of the continuum robot with ultrasound imaging in a phantom aortic arch. a** Schematic illustration shows that the robot can create a channel through the arteriae radialis to the common carotid artery for cerebral vascular intervention. **b** The robot follows a planned path with little environmental interaction (top). At the same time, the probe repeatedly scans along this path (middle) to reconstruct the overall shape of the robot in the vascular network (bottom). Scale bars, 20 mm.

generated by a permanent magnet to form a hook in situ in the porcine stomach with dimensions larger than the diameter of the cardia, which can be used as a surgical tool. Following in vivo experiments, the upper gastrointestinal tract of the live porcine model was examined using a commercial gastroscope (EV-230, Shenda Endoscope Co.) to evaluate the damage caused by this robot (Fig. S11). No obvious tissue or mucosal damage was found.

## Clinical safety and biocompatibility assessment

Although we initially assessed tissue damage qualitatively in the porcine esophagus, we further conducted in vitro experiments to quantitatively compare our robot's lateral force with that of a standard commercial catheter (ID = 3.5 mm, OD = 4.9, Fresenius Kabi India Pvt. Ltd.) in a large angular position (Fig. S12a, b, d, and Supplementary Movie 14). While our continuum robot can navigate a 180° bend without sidewall contact under precise positional control, we specifically tested the worst-case scenario where the robot remains close to the sidewall. The results show that the force exerted by our robot on the sidewall is nearly 100 times lower than that of the commercial catheter. Furthermore, we compared our robot's sidewall force with an existing LMPA-based continuum robot with variable stiffness (Fig. S12b, c), where both PTCs were simultaneously softened, advanced, and the sidewall forces recorded. Our robot exerted 10–15 times less force than the conventional variable stiffness robot. Additionally, the phase transition minimizes the pressure of the softened PTC on the support PTC, hence reducing friction, further aided by the hydrogel lubrication layer. The continuum robot used in our study is 400 mm long, and throughout all experiments, we observed no issues with deployment due to frictional resistance. This length satisfies certain medical requirements. Moreover, as the Guider becomes flexible before extending in each motion cycle, due to the minimal flexural strength, it buckles under resistance, ensuring that tissue damage caused by excessive pressure and popping forward is avoided.

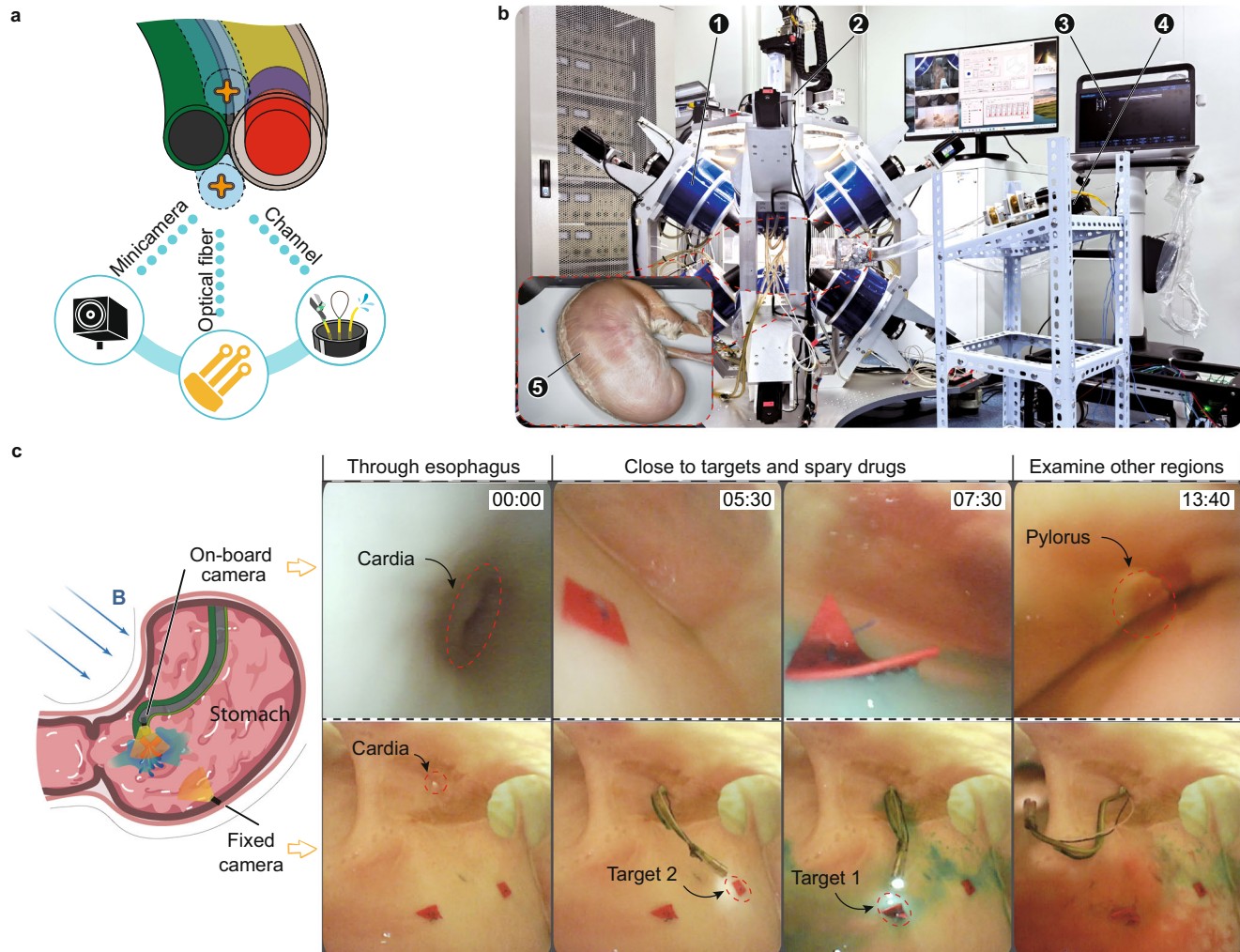

**Fig. 7 | Magnetic navigation of the continuum robot with endoscopic imaging for gastric treatment ex vivo. a** The robot can act as a motion unit with functional units attached. For example, equipped with minicameras, optical fibers, and manipulation channels, this robot can deliver drugs to stomach lesions. **b** The experimental environment of the gastric therapy on a porcine stomach. The numbered correspondence is (1) magnetic actuation system, (2) 4-DOF platform, (3) US imaging equipment, (4) advancement unit, and (5) porcine stomach. **c** The robot passed through the esophagus into the stomach. Then it found and reached stomach lesions with the help of an equipped minicamera. After spraying the drug around the lesion, another region was examined in the stomach. Images were taken by the on-board camera and a fixed camera in the porcine stomach.

Our experiments provide substantial evidence of the bio-compatibility of our robot. Initially, as demonstrated in Fig. S13, the PTC with its lubrication layer has been verified to be biocompatible. In the unlikely event of the lubrication layer being compromised, the PTC itself does not exhibit cytotoxic properties. Moreover, even in scenarios of potential leakage, the LMPA used in our robot does not show significant cytotoxicity. An additional layer of silicone tubing can be employed to ensure that any leaking LMPA remains isolated from the environment. Furthermore, any leakage would result in an increase in the electrical resistance of the PTC, as the cross-sectional area of the LMPA decreases. This change in resistance can be monitored to assess the state of the continuum robot (as shown in Fig. S14). It is also noteworthy that the design of our robot minimizes the presence of sharp edges that could potentially damage luminal structures. Additionally, the PTC's thermal management capability ensures that the surface temperature remains below 42°C in air (refer to Fig. S2d and Fig. S4b), which is even lower in liquid environments, thus keeping within the biocompatible temperature range[30]. In summary, these features enhance the clinical safety of our robot, making it a promising tool for medical applications.

## Discussion

The imperative need for small-scale soft continuum robots to autonomously extend apically while maintaining structural stability, without relying on forces generated by environmental contact, is paramount for enhancing surgical safety, reducing complexity, and expanding functional capabilities. In this study, we introduce a magnetic steering continuum robot exhibiting follow-the-leader (FTL) behavior to address this critical requirement. Our robot consists of two PTCs that can only move axially in relation to each other, alternately changing their stiffness and moving in periodic. Throughout each motion cycle, these components alternate between serving as a stable, solid-like backbone and a liquid-like element for progressive advancement. This design enables the robot's shape to be actively and dynamically programmed or reprogrammed through trajectory planning of its tip, thereby enabling operation independent of environmental interactions. This robot's structure, lubricating layer, working temperature, and control model have been optimized to satisfy the movement requirements. Combined with the clinical imaging technologies, it can low-invasively navigate along a planned path through a tortuous body lumen into a relatively open space of an organ, and then morph into functional structures in situ, which cannot be accomplished using

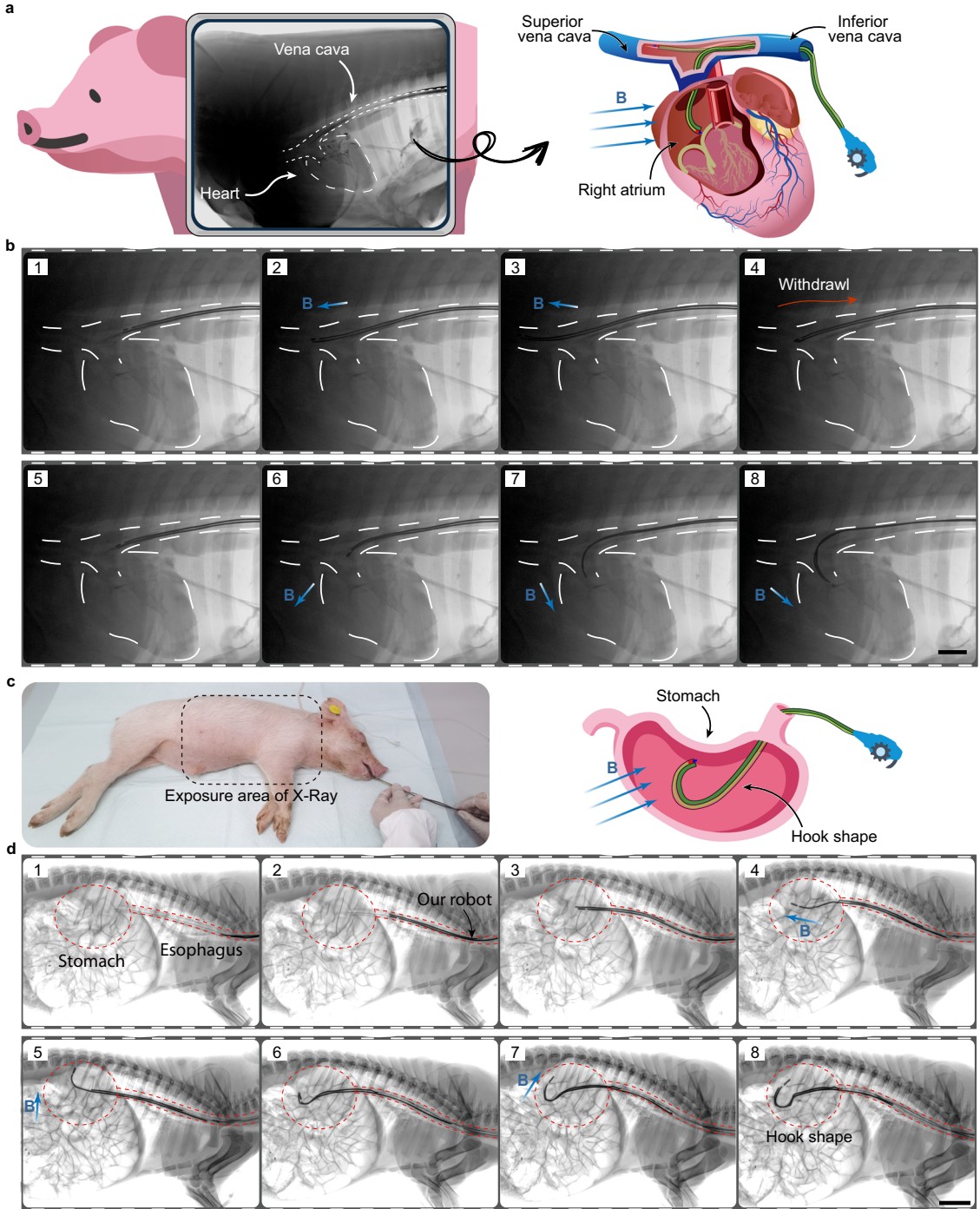

**Fig. 8 | In/ex vivo experiments assisted by X-ray imaging. a** Schematic of the robot passing through the inferior vena cava into the superior vena cava or the right atrium under magnetic navigation. The thorax of a porcine cadaver is imaged by X-ray after angiography. **b** Series of radiographic images document the deployment of the robot in the vena cava and heart. Scale bars, 30 mm. **c** The deployment of the robot in the porcine stomach in vivo. **d** Positioned by X-ray imaging, the robot is navigated through the esophagus into the porcine stomach and forms the shape of a hook in vivo under the applied magnetic field. Scale bars, 40 mm.

existing small-scale soft continuum robots[7–11,18–21,23,36,37]. The demonstrated capabilities in realistic, clinically relevant ex vivo and in vivo environments have illustrated the potential of the robot for medical applications.

Furthermore, integrating the robot with existing or cutting-edge medical technologies will promote clinical translation in the future. US- and X-ray-based imaging are well-established techniques regularly used for clinical diagnostic purposes. We have integrated ultrasound imaging into our system, and the experiment in the phantom vasculature showed that US, with the high spatial and temporal resolution,

provides real-time feedback on the overall position of the robot in deep tissue. Moreover, the experiments in the heart of a porcine cadaver and the stomach of a live porcine model exhibited that radiographic imaging can visually determine the in vivo environment as well as the location of the robot, so we will introduce X-ray imaging into our experimental system in future work. For more precise surgery in the complex environment in vivo, magnetic localization[38] will also be combined to obtain the tip's real-time attitude. The miniaturization of surgical instruments is also a proven technique for transluminal procedures. Depending on the working environment and the target,

several combinations of functional micro units can be loaded onto the robot. With images captured by a miniature camera, surgeons can operate microsurgical instruments to biopsy and procedure a lesion.

However, it should be noted that the large length-to-diameter ratio of structure and the small strength-to-density ratio of material result in a low flexural strength of our robot. Gravity deforms the robot, but this does not affect the robot's ability to perform transluminal procedures. First, although the robot is in localized contact with the environment due to gravity during navigation, the small pressure does not result in tissue damage or non-negligible frictional resistance. Second, although the functional structures formed by the robot are also intuitively affected by gravity, their small size makes the effect negligible. Moreover, magnetic force can be used to reduce the effect of gravity for high-precision operations.

In future clinical procedures, surgeons will use movable electromagnets[39] or commercially available magnetic navigation systems[40] to operate the magnetic steering continuum robot. Although the advanced hybrid control strategy will reduce the surgery time and lower the working temperature below ambient temperature[41], adding the miniature cooling structure is still challenging. In future work, we will develop corresponding preparation methods. We envisage a future where our improved robot can safely and rapidly pass through any lumen, carrying or in situ forming multiple surgical tools to accomplish clinical missions. This robot will enable a broader range of applications and a higher level of safety in transluminal robotic surgery. It will also provide more functional and intelligent tools for minimally invasive surgery.

## Methods

### Fabrication and structure analysis

The PTC was fabricated using commercially available materials, including Cerrolow 117 (Bolton), silicone tubes (Guofengyuan), a tiny permanent magnet, a resistive heater, silver wire, enameled wire, and polyvinylpyrrolidone hydrophilic coating solution (MediCoat, BioNational Biomaterials). The silicone tube's thickness was 0.15 mm. Two sizes of silicone tubes (ID = 1 mm, OD = 1.3 mm, and ID = 1.7 mm, OD = 2 mm) were used for encapsulation and assembly, respectively. The cylindrical permanent magnets with a diameter of 1.5 mm and a height of 8 mm had a magnetization strength of -0.8 MA/m. The resistivity of a resistive heater, silver wire, and enameled wire, all with a diameter of 0.1 mm, was $0.48 \times 10^{-6}$, $1.586 \times 10^{-8}$, and $1.7 \times 10^{-8}$ $\Omega \cdot m$ at 20 °C respectively.

Low melting point alloy (LMPA) was selected as the PTC's phase transition material because of a maximum elasticity modulus of several of GPa and the most excellent modulus ratio (maximum to minimum) among reported and analyzed state-of-the-art variable stiffness materials (Table S1, Fig. S15). Among them, Cerrolow 117 is used in medical applications due to the optimal phase transition temperature (47 °C)[42].

And interface phenomena (e.g., oxidation, wettability) of the liquid alloy hamper re-soldering between fractured surfaces. During fabrication, the silicone tube propped up by more injected LMPA maintained a continuous pressure on the LMPA after encapsulation. The internal pressure adds self-healing properties to the PTC, rejoining the two sides of the fracture when the PTC is heated (Fig. S1c). We describe the preparation process in detail in the Supplementary Materials. A universal tensile testing machine (strain rate 0.05%/s, ambient temperature 25°C) was used to test the mechanical properties of LMPA used in our robots (Fig. S16a). Although the ultimate stress is about 35 MPa, a strain of 40% was reached before the fracture occurred. This high level of ductility suggests that a continuum robot made of this material, when properly handled, is highly unlikely to fracture during clinical applications (Fig. S16b). To lock the 3D shape of the already deployed segment of the robot, the rigid PTC must have sufficient flexural strength to resist gravity and restrain the other flexible PTC, which constrains the dimension of the PTC. Two metrics were

calculated to represent the ability of a rigid PTC to maintain its shape (see calculation of PTC dimension in Materials and Methods for the exact procedure): (1) the maximum deflection of an 80 mm long robot supported by only one rigid PTC as a cantilever beam under gravity; and (2) the ratio of the flexural strength of a rigid PTC to that of a flexible one (RFS). The calculation shows that although the flexural strength of rigid PTCs of all diameters is much greater than that of flexible ones, the diameter still needs to exceed 1 mm to reduce the impact of gravity on the shape (Fig. 2c). However, if the effect of gravity on shape is not a concern, the radial dimensions of the PTCs can be further reduced for more minuscule and tortuous body lumina, such as cerebral vasculature. Due to the thickness of the silicone tubing encapsulating the LMPA, the ratio of flexural strength of the PTC in the rigid and flexible state reduces as the diameter decreases. To ensure the interlocking effect of the Follower and Guider, as well as the space required for the heating circuit, the diameter of the PTC should be larger than 0.5 mm.

### PTC heating circuit

As shown in Fig. S1a, the heating circuit contained a resistive heater, silver wire, LMPA, enameled wire, and a source meter. The resistive heater has high resistivity and a low temperature coefficient. When a current is loaded to the heating circuit, the energy is concentrated mainly on the resistive heater, and the temperature of LMPA increases due to the conduction effect of Joule heating released from the heating wire.

Silver wire is used to ensure the robustness of the heating circuit. When the rigid LMPA breaks, the contact of the fractured surfaces is so unstable that it will prevent the heating circuit from melting the LMPA. Therefore, the silver wire is connected in parallel with LMPA to hold the whole heating circuit closed at all times. Silver wire with a small cross-sectional area has a much higher resistance than the LMPA core so the total resistance $R_{tot}$ can be approximated as

$$R_{tot} = R_{Ew} + R_{Rh} + \frac{R_{LMPA}R_{Aw}}{R_{LMPA} + R_{Aw}} \approx R_{Ew} + R_{Rh} + R_{LMPA} \quad (1)$$

where $R_{Ew}$, $R_{Rh}$, $R_{LMPA}$, and $R_{Aw}$ represent the resistance of the enameled wire, resistive heater, LMPA, and silver wire, respectively. The terminals of the source meter were connected to the enameled wire, and the total resistance was detected while power was supplied. The mapping between temperature $T$ and the absolute change in total resistance $R_{tot}$ allows indirect detection of temperature. In addition, the change in LMPA's resistance in response to stimulation can be detected.

### Calculation of PTC dimension

The density, modulus of elasticity, thermal conductivity, and constant pressure heat capacity of the PTC's preparation materials were measured (Fig. S17). We considered the following scenario to evaluate the effect of gravity: a horizontally placed robot ($l = 80$ mm) with one end fixed, the Guider in a rigid state, and the Follower in a flexible state. Since only the rigid Guider maintains the shape of the robot, and the solid LMPA's elastic modulus $E_{LMPA}$ is much larger than that of the silicone tube, the maximum deflection of the robot $Y_g$ can be approximated as

$$Y_g = \frac{(m_G + m_F)gl^3}{8E_{LMPA}I_{LMPA-G}} \quad (2)$$

where $m_G$ and $m_F$ are the masses of the Guider and Follower, and the areal moment of inertia of the Guider's LMPA core with diameter $d$ can be calculated by $I_{LMPA-G} = \pi d^4 / 64$. We calculated the ratio of flexural strength RFS in this scenario as well. The rigid Guider ignores the effect of the silicone tube, while the flexible Follower ignores the impact of

the LMPA and treats the liquid LMPA as an empty space[29]:

$$RFS = \frac{E_{LMPA} I_{LMPA-G}}{E_{sili} I_{sili-F}} \qquad (3)$$

where $E_{sili}$ is the elastic modulus of the silicone tube and $I_{sili-F}$ is the area moment of inertia of the cross-section to the short axis of the Follower.

## Heat transfer simulation

Heat transfer simulation was implemented in the commercial finite element analysis software COMSOL. We performed steady-state and transient analysis using the 2D model in Figure S2a to obtain the current-to-steady-temperature and current-to-response-time relationships. The LMPA was encapsulated by the silicone tube and placed in a fluid environment. Joule heat was generated to heat the LMPA by current flowing through a resistive heater. Figure S2b shows the cross-section of the 3D model in the steady-state analysis. The areas highlighted in green and red respectively represented the Guider and Follower, surrounded by the fluid environment. After setting the temperature of the Guider or Follower, the other is heated due to heat conduction.

## Kinematic modeling

Based on the Euler-Bernoulli beam model and assuming a constant curvature[43], we can obtain the following analytical expression for inclination angle $\theta$ of the Guider's tip with length $l$ (see Supplementary Materials for detailed calculations):

$$\theta = \frac{lBm}{E_G^f I_G} \sin(\gamma - \theta) \qquad (4)$$

where $B$ and $m$ are the norms of the magnetic flux density and the magnetic moment, $\gamma$ represents the magnetic field inclination angle, $E_G^f$ is the equivalent Young's modulus of the flexible Guider, and $I_G$ is the area moment of inertia of the Guider. Equation 4 ignores the effect of gravity, but the long advance distance of the Guider in a single motion cycle will lead to a non-negligible effect of gravity, and the parameter interval in which the effect of gravity is neglected needs to be determined to control tip deformation accurately. A scenario was analyzed where the influence of gravity was most prominent: the flexible Guider with diverse protruding lengths was placed horizontally as a cantilever beam under both horizontal magnetic and vertical gravity fields. The deformation of the Guider in response to an actuation magnetic field was numerically simulated in the commercial finite element analysis software ABAQUS/Standard using the FEA model proposed by Zhao et al.[44,45], and the results were in good agreement with the experimental ones (Fig. S18). As the data summarized (Fig. 3b) shows, the Guider should only advance up to 40 mm in each motion cycle to minimize the interference of the gravitational field, where position errors within 10 mm are considered negligible. The 2D accessible area and tip inclination angle of the Guider's protruding segment ($l \in {}^{20,40}$ mm) at 30 mT magnetic flux density ($\gamma \in [0, \pi]$ rad) can be obtained from the solution of Eq. 4 (Fig. 3c). The 3D accessible area is simply a revolution of the 2D accessible area, which can be obtained by rotating the actuation fields around the axes. Despite the high agreement between theory and experiment (Fig. S19), the kinematic model has limitations because it does not consider the force generated by the magnetic gradient. Therefore, the magnetic field handle control has a higher priority in deployment.

## Experimental equipment

Unless otherwise specified, the above experiments were performed with the experimental equipment shown in Fig. 7b. The equipment comprises three parts: (1) a magnetic actuation system, eight amplifiers (HEA-200C, Nanjing Foneng Technology Industry Co., Ltd.), and a control computer for generating the magnetic field, (2) an ultrasonic imaging device (E2, SonoScape, Inc.), a 4-DOF platform, and two cameras (acA2040-120um, Basler, Inc.) for imaging, and (3) an advancement unit (Fig. S20) and source meter system (Keithley 2636B, Tektronix, Inc) for manipulating the robot. We built a magnetic actuation system for preliminary experiments in the laboratory. Spheres of 308 mm diameter could be placed in the device without mechanical collisions so that experimental animals, such as rats, rabbits, and pigs, could be accommodated. In practice, the robot was propelled by the advancement unit and deflected by the magnetic field generated by the magnetic actuation system. The ultrasound probe anchored to the 4-DOF platform enabled the localization in liquid environments. Since X-ray imaging equipment (IPET007, Shenzhen Bestcare Biotechnology Co., Ltd.) was not integrated into the above experimental system, the two in vivo experiments based on X-ray localization were not done in the system. As an alternative, we used a permanent magnet to generate the magnetic field and manually propel the robot forward.

## Magnetic navigation

The magnetic actuation system places eight electromagnets at the vertices of the square hexahedron along the diagonal. The device can generate a uniform magnetic flux density of up to 43 mT in any direction within a 10 cm long cubic workspace with <10% inhomogeneity. Magnetic flux density **B** is controlled by regulating current $I_i$ in each electromagnet. The electromagnets all work in a linear area. They are at a sufficient distance to avoid mutual influence so that magnetic flux density **B** can be calculated by the superposition law[46]:

$$\mathbf{B} = \sum \tilde{\mathbf{B}}_i I_i \qquad (5)$$

where $\tilde{\mathbf{B}}_i$ is the magnetic flux density generated per unit current excitation of the $i$-th electromagnet. After calibrating each electromagnet's ability to generate magnetic flux density $\tilde{\mathbf{B}}_i$, the current $I_i$ of each electromagnet can be calculated based on the required magnetic flux density **B**.

## Experiment in the stomach of a live porcine model

The animal model was a 15 kg male Large White pig fasted for 12 h before the experiment (Experimental Animal Center of Harbin Veterinary Research Institute). General anesthesia was administered by intramuscular injection of 0.4 ml of Zoletil 50, and the indwelling needle was placed into the ear margin vein. With the aid of a laryngoscope, a silicone tube coated with hydrogel was inserted into the esophagus. An air pump (Elveflow OB1 MK3 + ) with flow control inflated the upper gastrointestinal tract. The robot was then inserted through the silicone tube into the esophagus and through the cardia into the stomach. The moist environment in vivo activated the hydrogel on the surface, reducing mucosal damage caused by friction. The magnetic field generated by a permanent magnet was used for magnetic navigation in vivo. The shape formed by the robot in situ was continuously imaged by X-ray imaging. At the end of the experiment, the robot was withdrawn. After commercial gastroscopy, the pig was placed in a separate enclosure for awakening.

## Ex vivo cytotoxicity test

To evaluate the cytotoxicity of our continuum robot, live/dead cell staining was used to detect cell viability. The human umbilical vein endothelial HUVEC cells were cultured in Dulbecco's modified Eagle's medium (DMEM) (Gibco) supplemented with 10% (v/v) fetal bovine serum (Biological Industries). PTC with hydrogel layers, PTC without layers, and LMPA as cytotoxicity assay objects were used as cytotoxicity assay objects and placed in the culture environment of HUVEC cells, respectively. Cells cultured in medium only were used as positive

controls. All cells were grown at 37°C in the humidified incubator with 5% $CO_2$. Cells around the test object were taken every 24 h and stained with Calcein AM. The cells were washed once with PBS and observed under the fluorescence microscope by the excitation wavelength of 488 nm.

## Reporting summary

Further information on research design is available in the Nature Portfolio Reporting Summary linked to this article.

## Data availability

The data generated in this study are provided in the main text, supplementary information, and source data file. Additional data are available from the corresponding author on request. Source data are provided with this paper.

## Code availability

All the relevant code used to generate the results in this paper and the Supplementary Information is available upon request.

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

## Acknowledgements

We thank Prof. Ying Hu and Prof. Xingwen Wang from Life Science And Technology College, Harbin Institute of Technology for their great support with the cytotoxicity test. This work was supported in part by the National Natural Science Foundation of China under Grants 61925304 (H.X.), 62127810 (H.X.), and 62203138 (X.M.). All animals were used in accordance with the Guide for Care and Use of Laboratory Animals. Animal experiments were permitted by the Experimental Animal Welfare Ethics Committee of Harbin Institute of Technology (no.: IACUC-2023058).

## Author contributions

L.M., C.T., P.Y. and H.X. conceived and designed the study. L.M., C.T. and P.Y. designed the continuum robots and performed simulations and characterizations. P.Y. and L.M. developed methods of fabrication. L.M. developed kinematic modeling and C.T. performed the finite-element analysis. L.M., C.T. and X.S. developed the experimental equipment and its control interface. L.M., Y.P. and C.T. conducted the ex vivo experiments and analyzed the results. Y.P., L.M., F.W., C.T., H.Z. and X.S. conducted the in vivo animal study and analyzed the results. L.M. wrote the manuscript with input from all authors. X.M. assisted in revising the manuscript. H.X., X.M. and H.Z. supervised the study.

## Competing interests

The authors declare that they have no competing interests.
