## [Peer Review File · Nature Communications]

Reviewers' comments:

Reviewer #1 (Remarks to the Author):

In the attached manuscript, authors present a millimeter-scale magnetic soft continuum robot that consists of two parts (i.e., solid-like back-bone and liquid-like component for forward advancement). The sequential change in the stiffness of these components based on phase transition allows a shape-locking mechanism while advancing soft continuum robot tip under magnetic steering. They thermally characterize this phase transition to control the switching mechanism and demonstrate different applications employing in-vitro phantom models as well as in-vivo animal experiments. Although the shape-locking mechanism has been employed in a smart way, this reviewer does not see sufficient novelty in the submitted work to be considered worthy of publication in Nature Communications.

The following are specific major comments regarding the presented work.

1. First of all, authors overlook the literature in the introduction, resulting in inadequate explanation of their novelty with respect to the current state-of-the-art. The variable stiffness continuum robots have been demonstrated in many previous publications, e.g., C. Chautems et al., *Adv. Intell. Syst.*, 2020, 2, 1900086; Y. Piskarev et al., *Adv. Funct. Mater.*, 2022, 32, 2107662; J. Lussi et al., *Adv. Sci.*, 2021, 8, 2101290; M. Mattmann et al., *Adv. Sci.*, 2022, 9, 2103277). In fact, the presented fabrication method and the control over the phase transition seem extremely similar to the previous literature. However, authors fail to discuss their novelty over these existing works (though it looks like they are aware of these publications as they have already cited some of them in the SI). It may behoove the authors to be clearer in their introduction and discussion what the differences/improvements are between their approach and previously reported ones. Especially, considering that they are using same LMPA material and control approach (J. Lussi et al., *Adv. Sci.*, 2021, 8, 2101290); as well as similar shape locking method for advancing the additional catheter inside a variable stiffness guiding catheter (M. Mattmann et al., *Adv. Sci.*, 2022, 9, 2103277).

2. The authors are keen to make the claim that their approach is based on a nature inspired “growth-driven” mechanism. However, what this reviewer understands from the presented work is that they principally describe to advance a magnetic catheter (soft continuum robot) by utilizing a shape-locking mechanism based on phase transition of LMPA. The nature’s growing behavior is originated from addition and self-assembly of monomers (or comportsments), and cannot be mimicked by solely advancing a catheter with shape-locking mechanism.

3. The presented continuum robot is composed of two cylindrical variable stiffness catheters localized next to each other in parallel configuration. Although this mechanism allows to accomplish various shapes in open spaces, due to the asymmetric extension of the effective cross-section, it will be problematic while operating in the confined vascular channels. Could authors comment on these? Moreover, proposed additional functionality (i.e., extensibility) of the device (Figure 7) with supplementary tools (camera, working channel, optical fiber etc.) would result in a further increase the effective diameter of the device.

A). In connection to this point, as the continuum robot curls in open space (or around an object), twisting of side-by-side lumens could hinder advancing of the soft robot. Did authors observe this situation in the magnetic navigation experiments?

4. In page 8 line 153-155, author says “The PTC can transform from a rigid into an utterly flexible state in approximately 5 seconds in air and 10 seconds in water. Stiffening ends about 15 s and 10 s after the start of cooling in air and water, respectively” If they are not using any active cooling, these cooling times seems a bit short considering the several millimeters thick cross-section of LMPA in the device. Especially, it is difficult to understand how they can achieve the active heating and passive cooling within the same time (i.e., 10 s in water case).

5. The diameter of soft continuum robots are a critical design parameter, especially when they are planned to use in medical applications. In Figure 3d, it seems like the two components of the continuum device are separated (top-view), resulting in an effective diameter of more than 5 mm, which is obviously much larger than the designed value. Could authors comment on this, and assess the possible risks that can be encountered in real medical application due to this phenomenon.

6. In page 13 line 234-235, authors claim that they added the capability of segmental variable stiffness to the robot. To have different segments with actively controlled stiffness along the length of continuum robot is a very significant concept. It behooves the authors to give more detail on this fabrication process and show solid experimental characterizations where they can separately control the stiffness of each segment. For example, they easily demonstrate this under magnetic field bending via sequentially changing the stiffness of different segments.

7. Claiming that the knots formed along the continuum robots allows for in-situ pressure sensing requires more depth characterizations (Figure 5e). For example, what are the pressure/force levels that can be detected? What is the sensitivity of the proposed sensing mechanism based on the axial and radial directions? etc.

8. In the case of ex-vivo targeted drug delivery demonstration, it seems like authors don't have enough precision and accuracy while delivering the specific amount of the drug at selected target (Movie S9). Could they provide more quantitative information to evaluate the performance for targeted drug delivery.

9. In page 18 line 332-334, authors say "Following in vivo experiments, the upper gastrointestinal tract of the live porcine model was examined using a commercial gastroscope (EV-230, Shenda Endoscope Co.) to evaluate the damage caused by this robot (Fig. S6). No obvious tissue or mucosal damage was found." Can they provide any quantitative report such as histological evaluation of the tissue interacted with the continuum robot? Otherwise presented endoscope images (Fig. S6) are just qualitative and insufficient to comment on the tissue damage.

The following are some minor comments regarding the presented work.

1. Could author be more specific about the soft continuum robot's dimensions, especially regarding the cross section (effective diameter). They are providing some dimensions for different specific components (e.g., guider, follower, working and camera channels etc...) but I assume that assembling them together will result in a larger cross section since they add all components side-by-side.

2. Did authors characterize the force result in breaking of the continuum robot (Figure S1c)? knowing this value is important for minimizing the risks can be encountered during operation of the continuum robot.

3. Authors are claiming that there is no interaction with the surrounding walls (except the gravitational force related cases). This claim is hard to accept by looking at Fig. 5d and Movie S5 where the continuum robot interacts with the side walls of the maze. Could authors comment on this.

Reviewer #2 (Remarks to the Author):

The manuscript proposes a continuum robot comprising two 'phase transition components' (PTCs), namely flexible devices that alternate between solid and liquid state, based on a low melting point alloy (LMPA) encased in a silicone tube. One PTC, named Guider, is equipped with a tip magnet, to be steered by an external magnetic field for controlled incremental "growth" of the robot along a predefined path. The other PTC, called Follower, is also glued to a larger silicone tube within which the Guider slides. Both PTCs are coated with hydrogels in order to reduce friction during sliding. Resistive heaters are inserted within each PTC: LMPA phase transition is controlled via the flowing current (and temperature is sensed based on a calibrated resistive sensor in the same heating circuit). The proposed robot is devised for delicate medical intervention, where contacts with delicate vessel/organ walls have to be avoided (or at least minimized). A number of demonstrations are provided, ranging from tabletop experiments to ex vivo and in vivo tests.

Overall, the addressed biomedical problem is of clear relevance (it is an open problem, indeed), and the proposed scientific contribution features several point of interest. Some aspects, however, are not fully clear and need to be carefully improved, based on the following comments.

1. The robot was said to grow, and analogy was made to natural vines, yet no material is added at the tip of the proposed system. Rather, there is an incremental relative sliding between the two PTCs, which are individually pushed from the base. Hence, the call for a "growing robot" should be carefully reconsidered throughout the entire manuscript. Instead, the proposed system appears to align more closely with the concept of "follow the leader." Consequently, it would be advisable to conduct an analysis and comparison with the related state of the art. Furthermore, it's worth noting that the system is pushed from the base, and there is no actual growth occurring from the tip. This leads to an increase in the pressure applied by the robot during its movements. It would be interesting to gain more insight into the pressure generated and the potential tissue damage caused by deformation and stretching.

2. Related to the previous point, the parallelism between the continuum robot and the natural vine is not clear, including the comparison between the attraction towards light and an external magnetic field (Fig. 1a). In the former case, light serves as sensory feedback, while in the latter, a magnetic field is involved.

3. The paper neglected a closely related technology, similarly based on two flexible devices supporting each other in an alternate fashion based on stiffness switching. Some related manuscripts are provided below as examples: Choset, Howie, and Wade Henning. "A follow-the-leader approach to serpentine robot motion planning." *Journal of Aerospace Engineering* 12.2 (1999): 65-73; Gilbert, Hunter B., Joseph

Neimat, and Robert J. Webster. "Concentric tube robots as steerable needles: Achieving follow-the-leader deployment." *IEEE Transactions on Robotics* 31.2 (2015): 246-258; Kang, Byungjeon, Risto Kojcev, and Edoardo Sinibaldi. "The first interlaced continuum robot, devised to intrinsically follow the leader." *PloS one* 11.2 (2016): e0150278; Gao, Yuanqian, et al. "Continuum robot with follow-the-leader motion for endoscopic third ventriculostomy and tumor biopsy." *IEEE Transactions on Biomedical Engineering* 67.2 (2019): 379-390.

It is necessary to properly frame the proposed robot relative to close prior art: the introduction and discussion should be consistently revised.

4. It is not fully clear how the robot was knotted: how was the magnetic field gradient produced? How large was such a gradient? It seems that quite a large gradient would be needed to achieve the knotting reported through main figures and SI. The ms should consistently reconsider whether the knotting capability could be practically implemented.

5. The actuation unit (i.e., that one pushing the PTCs) should be further detailed (and possibly shown).

6. The operating temperatures should be clearly reported. Indeed, considering the thermal strategy at the core of the proposed robot, it is essential to clearly state which is the temperature effectively reached by the PTCs during their operation. If the current working temperatures are relatively high for clinical applications (as the mentioned 47deg could be), the associated limitation should be clearly stated. (A complementary recording with a thermal camera might add value.)

7. Some sentences are hard to understand (e.g., on line 166, "The advance distance should also not be less than 20 mm to balance the efficiency of the movement."). Subplots e and f in fig 5 seem to be switched. In the title of Section 2.8, both "ex vivo" and "in vivo" should be mentioned. Some typos should be amended.

Reviewer #3 (Remarks to the Author):

In this manuscript, the authors introduce a novel continuum robot inspired by biological growth mechanisms, enabling apical extension while maintaining structural stability. Guided by programmable magnetic fields and incorporating phase transition components, the proposed robot can autonomously undergo cycles of tip-based elongation, featuring a solid-like backbone for stability and a liquid-like component for controlled advancement. The authors then demonstrate the robot's capability in navigating through complex luminal environments and thus implying its biomedical applications in in vivo imaging. The article is well written ensuring a fluent read, and the figures are well drawn. In light of this, the reviewer recommends the article be accepted for publication in Nature Communications before addressing the comments below:

1. Can the authors benchmark the specifications of the proposed robot, such as maximum length, diameter, angulation range, and motion speed, with state-of-the-art commercial endoscopes? A comparative analysis of these parameters with other reported technologies and existing commercial endoscopes should be provided. Any clear limitations of this system should be discussed with potential solutions.
2. The reviewer finds the current Guide-Follower design quite cumbersome and space-occupying that may not be ideal for device fabrications as well as for operation under tight space such as in blood vessels. Can the authors discuss if there are any ways to integrate these two components and save space? For example can multilayer concentric tubes be used? Also, when integrating with fiber optics, can the fibers be inserted into the robot to further reduce the overall diameters?
3. The authors should do a chronic leakage test under simulated physiological environments and provide quantitative data to show the successful sealing of cytotoxic components.
4. What is the size limitation of this approach i.e. how small can it go? What are the associated challenges and potential solutions?
5. Although the robot does not require touching the lumina to move, it seems from Figure 6-8 that during actual in vivo operations the device inevitably contacts the tissues, which may eliminate the claimed advantages such as reduced potential medical risks and restricted mobility. The authors should discuss strategies to ensure little or no contact of the robot with the lumina throughout the in vivo operations.

Responses to Comments on “NCOMMS-23-43861-T”

We sincerely thank three reviewers for thoroughly reading our manuscript. And your constructive comments and suggestions are fully addressed in this letter point-by-point. For the convenience of the reviewers, the corresponding modifications are highlighted in purple font in this letter and with yellow background in the revised version. We believe that the questions raised by the reviewers have highlighted areas in need of further attention and the modifications have contributed to a significant improvement of our manuscript.

Response to comments from Reviewer #1

I. General Comments:

In the attached manuscript, authors present a millimeter-scale magnetic soft continuum robot that consists of two parts (i.e., solid-like back-bone and liquid-like component for forward advancement). The sequential change in the stiffness of these components based on phase transition allows a shape-locking mechanism while advancing soft continuum robot tip under magnetic steering. They thermally characterize this phase transition to control the switching mechanism and demonstrate different applications employing in-vitro phantom models as well as in-vivo animal experiments. Although the shape-locking mechanism has been employed in a smart way, this reviewer does not see sufficient novelty in the submitted work to be considered worthy of publication in Nature Communications.

■ Response:

We appreciate the reviewer's feedback regarding the novelty of our work. We aim to demonstrate through the arguments and new data presented in this response letter that our work indeed offers novel contributions. To the best of our knowledge, our research is among the first to report on small-scale continuum robots capable of growth-like motions along arbitrary paths. This was achieved using two variable stiffness continuum robots to facilitate shape locking. We carefully tuned the dimensions, temperature, and lubrication to enable this unique capability. Additionally, our work introduces the innovative concept of in situ formation of surgical instruments using a continuum robot, which we believe has not been previously explored in the literature. We are hopeful that our technique will significantly enhance safety and broaden the application spectrum for continuum robots. We have diligently worked to address the specific comments raised by the reviewer.

II. Major issues:

■ Comments #1:

First of all, authors overlook the literature in the introduction, resulting in inadequate explanation of their novelty with respect to the current state-of-the-art. The variable stiffness continuum robots have been demonstrated in many previous publications, e.g., C. Chautems et al., *Adv. Intell. Syst.*, 2020,2, 1900086; Y. Piskarev et al., *Adv. Funct. Mater.*, 2022, 32, 2107662; J. Lussi et al., *Adv. Sci.*, 2021, 8, 2101290; M. Mattmann et al., *Adv. Sci.*,2022, 9, 2103277). In fact, the presented fabrication method and the control over the phase transition seem extremely similar to the previous literature. However, authors fail to discuss their novelty over these existing works (though it looks like they are aware of these publications as they have already cited some of them in the SI). It may behoove the authors to be clearer in their introduction and discussion what the differences/improvements are between their approach and previously reported ones. Especially, considering that they are using same LMPA material and control approach (J. Lussi et al.,

Adv. Sci., 2021, 8, 2101290); as well as similar shape locking method for advancing the additional catheter inside a variable stiffness guiding catheter (M. Mattmann et al., Adv. Sci., 2022,9, 2103277).

■ Response #1:

We thank the reviewers for their valuable comments and apologize for the literature omission due to our negligence in writing the manuscript. We have made a summary of variable stiffness continuum robots in the revised manuscript (**Line 36 to 40, Page 2**). Nelson et al have made pivotal contributions in this field. We refer to some of these techniques and cite them in more detail in the revised manuscript (**Line 94, Page 5, Line 163, Page 9**).

To address reviewers' concerns about the novelty of our work, we compare the differences between our robot and the conventional variable stiffness continuous robots in **Table 1**. The variable stiffness magnetic continuum robot has the advantages of small scale, flexibility, and accessibility. It can reduce stiffness during deployment to minimize forces on tissues, but the forces still accumulate as the robot continues to deepen the lumen. This not only results in risks but also leads to localized buckling, preventing the transfer of thrust from proximal to distal. What's more, the similar shape-locking method mentioned by the reviewer (M. Mattmann et al., Adv. Sci., 2022,9, 2103277) is not interlocking but rather a process of delivering surgical tools through the working channel. Our proposed continuum robot, capable of growth-like motions along arbitrary paths, can safely navigate through the lumen without reliance on environmental interactions and form various functional structures in situ without pre-programming.

In response to the reviewer's suggestion to compare our approach with previously reported ones for variable stiffness, it is important to emphasize that our primary focus is on developing continuum robots capable of emulating growth-like motion, rather than solely concentrating on variable stiffness. Building upon the variable stiffness robot concept, we have delved deeper into a cooperative strategy involving two variable stiffness units. Our proposed design strategy for continuum robots, based on phase change elements, allows for growth-like motions: the two units are designed to interlock while ensuring their stiffness transitions are independent and allow for relative movement. To this end, we have implemented several key design features:

1. Structural analysis of the diameter range for mutual shape-locking (**Fig. 2c**);
2. Temperature control techniques to mitigate mutual influence between phase-change units (**Fig. 2d-g**);
3. Application of a lubrication layer to minimize friction between the units (**Fig. 2a**);
4. Incorporating cyclical motion to facilitate easier automated deployment of the robot (**Fig. 3**).

Additionally, extensive sections of our manuscript (**Fig. 4-8**) are dedicated to demonstrating the robot's enhanced motility, its ability to be functionalized, and its potential applications in the medical field, showcasing the capabilities of our robot with growth-like motion.

We added a summary of variable stiffness continuum robots to the revised manuscript and made comparisons to demonstrate the novelty of our work.

Line 36 to 40, Page 2 “The thermal response variable stiffness magnetic continuum robot has the advantages of small scale, flexibility, and accessibility (18-21). It can reduce stiffness during deployment to minimize forces on tissues, but the forces still accumulate as the robot continues to deepen the lumen. This not only results in the risks described above but also leads to localized buckling, preventing the transfer of thrust from proximal to distal.”

Line 74 to 79, Page 4

Table 1 Performance comparison of our robots with existing continuous robots

	Ref.	Actuation strategies	Diameter (mm)	Length (mm)	Angulation range (°)	Variable stiffness	Growth-like (FTL) deployment	Functionality	Application Scope	In vivo validation
Variable Stiffness Robot	Kim [10]	Magnetic	0.8	~200	±161	NO	NO	Carrying tools	Vascular intervention	Yes
	Lussi [18]	Magnetic	1	30	~ ±180	Thermal response	NO	Carrying tools	Ophthalmic Surgery	NO
	Mattmann [19]	Magnetic	1.3	140	-	Thermal response	NO	Carrying tools	Stomach and vessels	NO
	Chautems [20]	Magnetic	2.33	50	~ ±180	Thermal response	NO	Carrying tools	Vascular intervention	NO
	Piskarev [21]	Magnetic	2	88	±51	Thermal response	NO	Carrying tools	Vascular intervention	NO
FTL Robot	Amanov [31]	Tendon-driven	7	165	-	NO	Limited path	-	-	NO
	Gao [28]	Tendon-driven	3.4	60	~ ±180	NO	Limited path	-	Brain surgery	NO
	Gilbert [27]	Concentric tube	2.18	120	-	NO	Limited and predefined path	-	Brain surgery	NO
	Peyron [29]	Concentric tube+ Mag	1.07	213.5	±136.4	NO	Limited and predefined path	-	Nasal cavities and ear canals	NO
	Kang [30]	Multi-backbone	30	<200	±90	Mechanical locking	Arbitrary path	-	-	NO
This work	Magnetic	< 4	>400	~ ±180	Thermal response	Arbitrary path	Formation of functional structure or carrying tools	Vascular intervention and gastrointestinal surgery	Yes	
Commercial Catheters	Video cholangioscope (Innovex Medical Co.)		5	380	± 180	NO	NO	Carrying tools	Gastrointestinal tract	Yes
	Video bronchoscope BF-1TH190 (Olympus Co.)		6.2	600	± 120	NO	NO	Carrying tools	Bronchial tube	Yes
	Fiber pharyngoscope FS2 (Orlvision Co.)		2.9	300	± 130	NO	NO	Carrying tools	Nasopharynx	Yes
	Video cysto-nephroscope CYF-VH (Olympus Co.)		5.5	380	-130~220	NO	NO	Carrying tools	Urology	Yes

Compared with existing continuum robots, our robot, with small scale and flexibility, can achieve growth-like deployment along arbitrary paths. In addition to carrying tools, our robot can also form itself into functional structures in situ. The diameter, length, and angulation range of our robot have reached the level of commercial endoscopes. Through experiments, we found that our robot exerts 100 times and 10-15 times less force on the sidewall compared to commercial catheters and conventional variable stiffness robots, respectively.

■ **Comments #2:**

The authors are keen to make the claim that their approach is based on a nature inspired “growth-driven” mechanism. However, what this reviewer understands from the presented work is that they principally describe to advance a magnetic catheter (soft continuum robot) by utilizing a shape-locking mechanism based on phase transition of LMPA. The nature’s growing behavior is originated from addition and self-assembly of monomers (or compartments), and cannot be mimicked by solely advancing a catheter with a shape-locking mechanism.

■ Response #2:

We apologize for the ambiguity due to imprecise expressions and acknowledge Reviewer #1's observation regarding the distinction between our robot's mechanism and natural growth processes. While the underlying principle of our continuum robot's movement differs from that of natural growth in vines, the outcome mirrors aspects of growth, notably in maintaining the shape of deployed segments and navigating complex environments without dependency on interactive forces. Considering this, we have revised our terminology from 'growth-driven' to 'growth-inspired' in the manuscript. This change aims to provide a clearer and more accurate depiction of our robot's functionality, aligning the description with the actual motion mechanism while still acknowledging the inspiration drawn from natural growth behaviors.

■ Comments #3:

The presented continuum robot is composed of two cylindrical variable stiffness catheters localized next to each other in parallel configuration. Although this mechanism allows to accomplish various shapes in open spaces, due to the asymmetric extension of the effective cross-section, it will be problematic while operating in the confined vascular channels. Could authors comment on these? Moreover, proposed additional functionality (i.e., extensibility) of the device (Figure 7) with supplementary tools (camera, working channel, optical fiber etc.) would result in a further increase the effective diameter of the device.

A). In connection to this point, as the continuum robot curls in open space (or around an object), twisting of side-by-side lumens could hinder advancing of the soft robot. Did authors observe this situation in the magnetic navigation experiments?

■ Response #3:

We recognize the concerns regarding the forces between the Guider and Follower, particularly the twisting from asymmetric extension during curling. However, this does not impede the relative sliding of the two PTCs, as the hydrogel significantly reduces friction. Consequently, our continuum robots can navigate blood vessels more safely and form diverse shapes in open spaces. It is important to note, though, that due to the small diameter of blood vessels, creating various shapes within them is limited by the design dimensions of our robot. We describe the influence of distortion in the revised manuscript (**Line 204 to 207, Page 10**) "As can be seen in the video, although the forces between the Guider and Follower increase, due to the twisting caused by the asymmetric extension of the effective cross-section when curling, this does not hinder the relative sliding of the two PTCs, since the hydrogel considerably reduces the coefficient of friction."

We explain the detailed dimensions of the continuum robot (**Line 112 to 114, Page 5**). "The maximum size of the cross-section of the robot is about 3-4 mm. Its length can be up to meters, here we use a robot with a length of about 400 mm for demonstrations (**Fig. S1d**)."

Regarding the integration of supplementary tools, as shown in **Fig. S10a** and **S10b**, these are incorporated at the Guider's tip, thus maintaining the overall compact size of the robot. We have added detailed dimensions in the revised version (**Line 323 to 327, Page 18**). "As an example relevant to potential medical applications, we equipped the robot with a working channel (ID = 0.5 mm) and a minicamera with built-in illumination (OD = 1.2 mm) to demonstrate the concept of diagnosis and treatment of gastric lesions (**Fig. 7a** and **Fig. S10a**). The supplementary tools are loaded in the Guider's tip, which keeps the overall size from increasing (**Fig. S10b**)."

Figure S1. Construction and self-healing of phase transition component (PTC). (a) Internal heating circuit of PTC. Energy is mainly consumed by the resistive heater. The temperature is monitored indirectly by the detection of LMPA resistance. When LMPA breaks, the silver wire preserves the heating circuit intact. (b) End view of PTC in various sizes. PTC size can be further reduced if the mission is confined to a surface where gravity does not disturb the shape. (c) Self-healing property of PTC. Due to the continuous pressure exerted by the silicone tube on the LMPA, the fractured PTC can overcome the LMPA's interface phenomena at the fracture surface after melting, thereby achieving self-healing. (d) Overall, partial and sectional views of the robot.

Figure S10. Robot equipped with endoscopic tools. (a) Overall, partial and sectional views of the robot with supplementary tools (a working channel and a minicamera with built-in illumination). (b) The supplementary tools are loaded in the Guider's tip, which keeps the overall size from increasing. (c) In the clinic, another growth-like robot can be used to arrange the minicamera for global images. (d) Relationship between protrusion length of the guider and its tip displacement under a magnetic field. The confined space inside the body limits the length of the deployed robot, so the magnetic field has little effect on the rigid robot

■ **Comments #4:**

In page 8 line 153-155, author says “The PTC can transform from a rigid into an utterly flexible state in approximately 5 seconds in air and 10 seconds in water. Stiffening ends about 15 s and 10 s after the start of cooling in air and water, respectively” If they are not using any active cooling, these cooling times seems a bit short considering the several millimeters thick cross-section of LMPA in the device. Especially, it is difficult to understand how they can achieve the active heating and passive cooling within the same time (i.e., 10 s in water case).

■ **Response #4:**

We appreciate this critical question and apologize for not discussing warming and cooling times in detail in the manuscript. As can be seen in **Fig. 2f**, the LMPA we used can be completely stiffened by decreasing the temperature from 49.4°C to 47°C. Moreover, the diameter of the PTC is slightly larger than 1 mm. It is thus reasonable that the phase transition is completed in air and water in 15s and 10s. We did not intentionally make the heating time and cooling time of the PTC in water the same. When cooling, the high thermal conductivity of water allows the PTC to harden completely in about 10s. When heating, the PID extends the warming time. Theoretically, the heating time in water does not exceed 3s (**Fig. 2d**). To provide a more comprehensive understanding, we have included additional details in the manuscript (**Lines 172 to 176**) about the stiffening process, which completes approximately 15 seconds in air and 10 seconds in water after the start of cooling (as shown in **Fig. S3 d, e** and **Movie S1**). The simulations (**Fig. S3**

a, b, c) support the feasibility of a quick transition from a flexible to a rigid state in PTC. The variances noted between experimental and simulated cooling times may be due to the robot's preparation variances and the environmental conditions during the experiments.

To address this issue, we have added the following specific changes in **Line 172 to 176, Page 9** “Stiffening ends about 15 s and 10 s after the start of cooling in air and water, respectively (**Fig. S3 d, e and Movie S1**). As can be seen from the heat transfer simulation ((**Fig. S3 a, b, c**), theoretically, a completely flexible PTC can cool down to a completely rigid state quickly. The difference between the experiment and the simulation may lie in the deviation of the robot preparation as well as the difference in the experimental environment.”

Figure S3. Heating and cooling of PTC. (a) Cooling Simulation of PTC. (b) Simulation results of PTC’s cooling time in water. (c) Simulation results of PTC’s cooling time in water. (d) and (e) First, both PTCs are heated simultaneously, and after about 6s (in air) and 10s (in water) they become completely flexible and bend downward under the force of gravity. Then, both PTCs were bent upward in response to the applied magnetic field. At the same time the heating of one PTC stops and it cools naturally in the environment. After about 16s (in air) and 10s (in water), the magnetic field was withdrawn and the heated PTC sagged under gravity, while the cooled PTC kept its shape unchanged, which was seen to become completely rigid. Scale bars, 10 mm.

■ **Comments #5:**

The diameter of soft continuum robots are a critical design parameter, especially when they are planned to use in medical applications. In Figure 3d, it seems like the two components of the continuum device are separated (top-view), resulting in an effective diameter of more than 5 mm, which is obviously much larger than the designed value. Could authors comment on this, and assess the possible risks that can be encountered in real medical application due to this phenomenon.

■ Response #5:

We appreciate the scrutiny of the reviewers. Upon re-examination, we confirmed the structural integrity of the robot, ensuring that the two components were indeed intact. The adhesive used exhibits tensile and shear strengths exceeding 1 MPa, fulfilling the requirements for securing soft PTCs. In the unlikely event that a small segment becomes detached during actual medical applications, it should not significantly impact the robot's kinematic performance due to the stability provided by the remaining segments. The observed increase in effective diameter is primarily attributed to the current proof-of-concept stage of our robot, where manual assembly may introduce some dimensional variances. Future development includes the implementation of automated manufacturing processes to maintain consistent axial dimensions of the robot's cross-section. Additionally, employing '8' shaped silicone tubing as an alternative could ensure uniform dimensions consistently.

■ Comments #6:

In page 13 line 234-235, authors claim that they added the capability of segmental variable stiffness to the robot. To have different segments with actively controlled stiffness along the length of continuum robot is a very significant concept. It behooves the authors to give more detail on this fabrication process and show solid experimental characterizations where they can separately control the stiffness of each segment. For example, they easily demonstrate this under magnetic field bending via sequentially changing the stiffness of different segments.

■ Response #6:

We appreciate the reviewer's comments, which enable us to better present our work. We add fabrication details of the segmented variable stiffness as well as demonstration experiments in **Line 259 to 265, Page 14** “The internal heating circuit of the PTC is shown in **Fig. S4a**: an additional resistive heater was added to the circuit. By selecting the resistive heater, the length of the variable stiffness segment of the PTC can be adjusted. Thus the robot maintained the shape of the rigid anterior segment to capture the object when withdrawing (**Fig. 5d** and **Movie S6**). Moreover, the initial resistive heater had a segment folded back at the tip of the PTC, which allowed segment 1 to be heated with twice the power of segment 2 (**Fig. S4a**). The current can be adjusted to make segment 1 flexible while segment 2 remains rigid (**Fig. S4b** and **Movie S7**).”

Figure S4. PTC with segmented variable stiffness capability. (a) Internal heating circuit of PTC. An additional resistive heater is added. (b) First, the resistance heater 2 was heated to soften segment 2. Then the PTC was bent and cooled under magnetic torque. Next, a suitable current was passed to the heating resistive heater 1 so that segment 1 was flexible while segment 2 remained rigid. Finally, a magnetic field was applied to bend the tip of the PTC. Scale bars, 10 mm.

■ **Comments #7:**

Claiming that the knots formed along the continuum robots allows for in-situ pressure sensing requires more depth characterizations (Figure 5e). For example, what are the pressure/force levels that can be detected? What is the sensitivity of the proposed sensing mechanism based on the axial and radial directions? etc.

■ **Response #7:**

We hope that the following work will answer the reviewer's comment. We characterize the knot formed by continuum robots. We have now added the text below in **Line 275 to 277, Page 14** “We measured the sensitivity of the knot to the radial force (**Fig. S6a, b**). The applied force and the amount of resistance change were linear within the range of 0 - 0.4 N ($k = 41.24 \text{ m}\Omega/\text{N}$, $R^2 > 0.985$) (**Fig. S6c**).”

Figure S6. Characterization of the sensitivity of the knot to radial forces. (a) Measurement system. The system consisted of two mechanical stages, a force gauge (Futek 538493) for gradually ramping up force with precise control, and a sourcemeter (Keithley 2636B) for recording real-time resistive responses. (b) Real-time recording of the amount of resistance change and force. (c) Relationship between the amount of resistance change and radial force.

■ **Comments #8:**

In the case of ex-vivo targeted drug delivery demonstration, it seems like authors don't have enough precision and accuracy while delivering the specific amount of the drug at selected target (Movie S9). Could they provide more quantitative information to evaluate the performance for targeted drug delivery.

■ **Response #8:**

We thank the reviewer for their valuable comment on precision and accuracy. We quantified the precision and accuracy of the robot's movements. We have now added the text below in

Line 320 to 323, Page 17-18 “Meanwhile, we characterized the precision and accuracy of the continuum robot by testing its localization accuracy for reaching the target position several times ($n = 5$) in a row (**Fig. S9a** and **Movie S11**). **Figure S8b** demonstrated that with good visual feedback, the robot's localization accuracy was 0.791 ± 0.291 mm and the motion trajectory remained essentially unchanged over multiple deployments.”

Line 336 to 337, Page 18 “In practical medical applications flow control devices can be used to precisely control the amount of drug.”

Figure S9. Precision and accuracy testing of the continuum robot. (a) Localization accuracy of a continuum robot reaching the same position multiple times ($n = 5$) in a row. The error of the robot's tip to the target is represented in the embedded figure. Scale

bars, 10 mm. (b) Distribution of localization errors of the continuum robot.

■ **Comments #9:**

In page 18 line 332-334, authors say “Following in vivo experiments, the upper gastrointestinal tract of the live porcine model was examined using a commercial gastroscope (EV-230, Shenda Endoscope Co.) to evaluate the damage caused by this robot (Fig. S6). No obvious tissue or mucosal damage was found.” Can they provide any quantitative report such as histological evaluation of the tissue interacted with the continuum robot? Otherwise presented endoscope images (Fig. S6) are just qualitative and insufficient to comment on the tissue damage.

■ **Response #9:**

We apologize for not giving a quantitative analysis. We acknowledge the importance of clinical-grade in vivo experiments; however, such studies require extensive qualification, funding, and time, which are currently beyond the scope of our capabilities. While a histologic evaluation was not conducted, we have performed quantitative tests to assess the force exerted by our continuum robot on sidewalls, thereby demonstrating its safety. Furthermore, we have conducted cytotoxicity experiments to confirm the biocompatibility of our robot. This additional information has been included in the manuscript (**Lines 377 to 405, Page 20-21**). “Although we initially assessed tissue damage qualitatively in the porcine esophagus, we further conducted in vitro experiments to quantitatively compare our robot's lateral force with that of a standard commercial catheter (ID=3.5mm, OD=4.9, Fresenius Kabi India Pvt. Ltd.) in a large angular position (**Fig. S12a, b, d** and **Movie S14**). While our continuum robot can navigate a 180° bend without sidewall contact under precise positional control, we specifically tested the worst-case scenario where the robot remains close to the sidewall. The results show that the force exerted by our robot on the sidewall is nearly 100 times lower than that of the commercial catheter. Furthermore, we compared our robot's sidewall force with an existing LMPA-based continuum robot with variable stiffness (**Fig. S12b, c**), where both PTCs were simultaneously softened, advanced, and the sidewall forces recorded. Our robot exerted 10-15 times less force than the conventional variable stiffness robot. Additionally, the phase transition minimizes the pressure of the softened PTC on the support PTC, hence reducing friction, further aided by the hydrogel lubrication layer. The continuum robot used in our study is 400 mm long, and throughout all experiments, we observed no issues with deployment due to frictional resistance. This length satisfies certain medical requirements. Moreover, as the Guider becomes flexible before extending in each motion cycle, due to the minimal flexural strength, it buckles under resistance, ensuring that tissue damage caused by excessive pressure and popping forward is avoided.

Our experiments provide substantial evidence of the biocompatibility of our robot. Initially, as demonstrated in **Fig. S13**, the PTC with its lubrication layer has been verified to be biocompatible. In the unlikely event of the lubrication layer being compromised, the PTC itself does not exhibit cytotoxic properties. Moreover, even in scenarios of potential leakage, the LMPA used in our robot does not show significant cytotoxicity. An additional layer of silicone tubing can be employed to ensure that any leaking LMPA remains isolated from the environment. Furthermore, any leakage would result in an increase in the electrical resistance of the PTC, as the cross-sectional area of the LMPA decreases. This change in resistance can be monitored to assess the state of the continuum robot (as shown in **Fig. S14**). It is also noteworthy that the design of our robot minimizes the presence of sharp edges that could potentially damage luminal structures. Additionally, the PTC's thermal management capability ensures that the surface temperature remains below 42°C in air (refer to **Fig. S2d** and **Fig. S4b**), which is even lower in liquid environments, thus keeping within the biocompatible temperature range. In summary, these features enhance the clinical safety of our robot, making it a promising tool for medical applications”

Figure S12. Force on the sidewall during a 180° turn. (a) Measuring the force on the sidewalls of a continuum robot as it moves forward. One side of the pressure sensor is fixed on the pedestal and the other side is connected to the corner sidewall. To avoid the effect of friction, the corner sidewall is not in contact with the pedestal. The force of the robot on the sidewall is captured by the sensor. (b) Pressure on the corner wall when two PTCs are alternately advanced. (c) Pressure on the corner wall when both PTCs are softened and advanced at the same time. (d) Pressure on corner wall when commercial catheter (OD=15Fr) is advanced. Scale bars, 10 mm.

Figure S13. Cytotoxicity Test of PTC and LMPA on HUVEC Cells. PTC with hydrogel layers, PTC without layers, and LMPA were placed in the culture environment of HUVEC cells for 3 days for cytotoxicity test. Scale bars, 100 μm .

Figure S14. Leakage detection in PTC. If the PTC is punctured, the pressure inside it will squeeze out the LMPA, causing the radial dimension to decrease and the resistance to rise. If an abnormal rise in resistance is detected, the continuum robot should be withdrawn quickly.

III. Minor issues:

■ Comments #10:

Could author be more specific about the soft continuum robot's dimensions, especially regarding the cross section (effective diameter). They are providing some dimensions for different specific components (e.g., guider, follower, working and camera channels etc...) but I assume that assembling them together will results in a larger cross section since they add all components side-by-side.

■ Response #10:

We have measured the dimensions of our continuum robot, including its cross-sectional area. For a detailed breakdown of these measurements, including the specific dimensions of components such as the guider, follower, and the channels for working and

camera, please refer to our response to **Major Issue #3**. This response provides comprehensive information on how the assembly of these components influences the overall effective diameter of the robot.

We explain the detailed dimensions of the continuum robot (**Line 112 to 114, Page 5**). “The maximum size of the cross-section of the robot is about 3-4 mm. Its length can be up to meters, here we use a robot with a length of about 400 mm for demonstrations (**Fig. S1d**).”

Regarding the integration of supplementary tools, as shown in **Fig. S10a** and **S10b**, these are incorporated at the Guider's tip, thus maintaining the overall compact size of the robot. We have added detailed dimensions in the revised version (**Line 323 to 327, Page 18**). “As an example relevant to potential medical applications, we equipped the robot with a working channel (ID = 0.5 mm) and a minicamera with built-in illumination (OD = 1.2 mm) to demonstrate the concept of diagnosis and treatment of gastric lesions (**Fig. 7a** and **Fig. S10a**). The supplementary tools are loaded in the Guider's tip, which keeps the overall size from increasing (**Fig. S10b**).”

■ Comments #11:

Did authors characterize the force result in breaking of the continuum robot (Figure S1c)? knowing this value is important for minimizing the risks can be encountered during operation of the continuum robot.

■ Response #11:

We acknowledge that the fracture force varies depending on the robot's size and the specific position of stress. To address this concern, we conducted tests on the mechanical properties of the LMPA used in our robots. As elaborated in **Lines 478 to 482, Page 25** of our manuscript, “A universal tensile testing machine (strain rate 0.05%/s, ambient temperature 25°C) was used to test the mechanical properties of LMPA used in our robots (**Fig. S16a**). Although the ultimate stress is about 35 MPa, a strain of 40% was reached before the fracture occurred. This high level of ductility suggests that a continuum robot made of this material, when properly handled, is highly unlikely to fracture during clinical applications (**Fig. S16b**).” If an unexpected fracture occurs, repair can also be accomplished by heating the LMPA.

Figure S16. Mechanical properties testing of LMPA (Bolton 117). (a) Measurement system (IBTC-5000, CARE Measurement & Control Co., Ltd., China). (b) Stress-strain curve of LMPA

■ Comments #12:

Authors are claiming that there is no interaction with the surrounding walls (except the gravitational force related cases). This claim is hard to accept by looking at Fig. 5d and Movie S5 where the continuum robot interacts with the side walls of the maze. Could authors comment on this.

■ Response #12:

We thank the reviewers for their concerns about our robot's contact with the environment. While it is true that our robot may occasionally come into contact with surrounding environments during deployment due to control precision or environmental factors, it's important to note that these interactions are not relied upon for the robot's deformation. We conducted tests to measure the force exerted by our robot on the sidewall during such contact, and the results confirm that the force exerted by our robot on the sidewall is nearly 100 times lower than that of the commercial catheter and 10-15 times lower than that of the conventional variable stiffness robot (**Fig. S12**). Therefore, the contact of our robot with its environment does not constitute a safety risk. Enhancing the accuracy of positioning and reducing the length of the protruding segment in each cycle can further improve the control accuracy of the magnetic field, thereby minimizing the robot's contact with its surroundings. Additionally, dynamic adjustment of control parameters during deployment helps the robot adapt to complex environments and avoid unnecessary contact. Regarding the scenario in Figure 5d, we opted to soften and withdraw the robot as a whole for demonstration purposes, showcasing the robot's capability for rapid retraction in certain situations. Although maneuvering the overall flexible robot exerts pressure on the sidewalls, the robot can be withdrawn quickly and the safety of withdrawal is higher than that of advancement. Moreover, it is also possible to retract the robot along the same path it deployed (as in **Fig. 4**), ensuring no contact with the environment, using the approach that the Guider's tip maintains the shape, and the remaining segments move alternately with the Follower.

Response to comments from Reviewer #2

I. General Comments:

The manuscript proposes a continuum robot comprising two ‘phase transition components’ (PTCs), namely flexible devices that alternate between solid and liquid state, based on a low melting point alloy (LMPA) encased in a silicone tube. One PTC, named Guider, is equipped with a tip magnet, to be steered by an external magnetic field for controlled incremental “growth” of the robot along a predefined path. The other PTC, called Follower, is also glued to a larger silicone tube within which the Guider slides. Both PTCs are coated with hydrogels in order to reduce friction during sliding. Resistive heaters are inserted within each PTC: LMPA phase transition is controlled via the flowing current (and temperature is sensed based on a calibrated resistive sensor in the same heating circuit). The proposed robot is devised for delicate medical intervention, where contacts with delicate vessel/organ walls have to be avoided (or at least minimized). A number of demonstrations are provided, ranging from tabletop experiments to ex vivo and in vivo tests.

Overall, the addressed biomedical problem is of clear relevance (it is an open problem, indeed), and the proposed scientific contribution features several point of interest. Some aspects, however, are not fully clear and need to be carefully improved, based on the following comments.

■ Response:

We appreciate the reviewer's recognition of the relevance and interest in our approach and findings. We are committed to providing further clarifications and have conducted additional experiments to address the concerns raised. We hope that these supplementary details and results will fully address the points mentioned by the reviewer.

II. Major issues:

■ Comments #1:

The robot was said to grow, and analogy was made to natural vines, yet no material is added at the tip of the proposed system. Rather, there is an incremental relative sliding between the two PTCs, which are individually pushed from the base. Hence, the call for a “growing robot” should be carefully reconsidered throughout the entire manuscript. Instead, the proposed system appears to align more closely with the concept of “follow the leader.” Consequently, it would be advisable to conduct an analysis and comparison with the related state of the art. Furthermore, it's worth noting that the system is pushed from the base, and there is no actual growth occurring from the tip. This leads to an increase in the pressure applied by the robot during its movements. It would be interesting to gain more insight into the pressure generated and the potential tissue damage caused by deformation and stretching.

■ Response #1:

1) We concur with Reviewer #2 that the motion principle of our continuum robots differs from natural vines' growth. However, the motion performance of our robot is akin to growth processes, as both can maintain the shape of deployed segments and navigate complex terrains without relying on interactive forces. To enhance clarity and accuracy in our manuscript, we have revised the term ‘growth-driven’ to ‘growth-inspired’ and toned down references to plant growth mechanisms.

2) Our system indeed aligns with the ‘follow the leader’ (FTL) approach, and our FTL continuum robot exhibits growth-like motion characteristics. In the Introduction, we provide an analysis of existing FTL continuum robots and compare their performances in **Table 1**. Unlike traditional FTL robots that are limited to predefined paths or lack flexibility, our robot is small-scaled, agile, and capable of FTL movement, thereby extending the functional scope of such robots. Similar to the diverse shapes achieved by vines, our robot can form various functional structures, offering dual utility as a tool carrier and a surgical instrument in medical procedures.

We have added the text and Table 1 below in the revised manuscript (Line 43 to 56, Page 2-3 and Line 75 to 80, Page 4). “Therefore, continuum robots should have the ability to actively comply with the features of the lumina and reach the target independently of tissue interaction. Through growth, bines can navigate complex terrains without relying on forces generated by interaction, which fulfills the above-mentioned requirement. Growing robots with similar growth principles to bines can retain the form of their previously grown segments while navigating (24,25), but suffer from large scale, low flexural strength, and long deployment times. Robots with the ‘follow the leader’ (FTL) behavior can also be deployed like bine growth. The FTL continuum robots are thus well-suited for navigation through the patient's anatomy while avoiding sensitive areas. FTL behavior is defined as the motion of a robot body along a path led by the tip (26) and can be implemented by concatenating several independent active deformation segments for more degrees of freedom (DOF). However, due to the difficulty of arranging deformation segments as compactly as muscles, these robots can only achieve FTL deployment along limited or predefined trajectories (27-29). Another FTL continuum robot is based on alternating deployment of two concentrically arranged snake robots with shape-lock mechanism (30), but the complex locking structure leads to large cross-section sizes and small angulation range.”

Table 1 Performance comparison of our robots with existing continuous robots

	Ref.	Actuation strategies	Diameter (mm)	Length (mm)	Angulation range (°)	Variable stiffness	Growth-like (FTL) deployment	Functionality	Application Scope	In vivo validation
Variable Stiffness Robot	Kim [10]	Magnetic	0.8	~200	±161	NO	NO	Carrying tools	Vascular intervention	Yes
	Lussi [18]	Magnetic	1	30	~ ±180	Thermal response	NO	Carrying tools	Ophthalmic Surgery	NO
	Mattmann [19]	Magnetic	1.3	140	-	Thermal response	NO	Carrying tools	Stomach and vessels	NO
	Chautems [20]	Magnetic	2.33	50	~ ±180	Thermal response	NO	Carrying tools	Vascular intervention	NO
	Piskarev [21]	Magnetic	2	88	±51	Thermal response	NO	Carrying tools	Vascular intervention	NO
FTL Robot	Amanov [31]	Tendon-driven	7	165	-	NO	Limited path	-	-	NO
	Gao [28]	Tendon-driven	3.4	60	~ ±180	NO	Limited path	-	Brain surgery	NO
	Gilbert [27]	Concentric tube	2.18	120	-	NO	Limited and predefined path	-	Brain surgery	NO
	Peyron [29]	Concentric tube+ Mag	1.07	213.5	±136.4	NO	Limited and predefined path	-	Nasal cavities and ear canals	NO
	Kang [30]	Multi-backbone	30	<200	±90	Mechanical locking	Arbitrary path	-	-	NO
This work	Magnetic	< 4	>400	~ ±180	Thermal response	Arbitrary path	Formation of functional structure or carrying tools	Vascular intervention and gastrointestinal surgery	Yes	
Commercial Catheters	Video cholangioscope (Innovex Medical Co.)		5	380	± 180	NO	NO	Carrying tools	Gastrointestinal tract	Yes
	Video bronchoscope BF-1TH190 (Olympus Co.)		6.2	600	± 120	NO	NO	Carrying tools	Bronchial tube	Yes
	Fiber pharyngoscope FS2 (Orlvision Co.)		2.9	300	± 130	NO	NO	Carrying tools	Nasopharynx	Yes
	Video cysto-nephroscope CYF-VH (Olympus Co.)		5.5	380	-130~220	NO	NO	Carrying tools	Urology	Yes

Compared with existing continuum robots, our robot, with small scale and flexibility, can achieve growth-like deployment along arbitrary paths. In addition to carrying tools, our robot can also form itself into functional structures in situ. The diameter, length, and angulation range of our robot have reached the level of commercial endoscopes. Through experiments, we found that our robot

exerts 100 times and 10-15 times less force on the sidewall compared to commercial catheters and conventional variable stiffness robots, respectively

3) The conventional catheters are passively flexed and depend on the force generated by interaction with luminal walls. Along with the movement, the thrust at the proximal end is increasing to overcome the rising friction force, potentially causing tissue damage. Our robot, although pushed from the base, applies significantly less pressure on tissues. First, our robot navigates without relay on environmental interaction. Our experiments demonstrate that the force exerted by our robot on the environment is almost 100 times less than that of commercial catheters, thereby greatly reducing the risk of tissue damage. Second, the phase transition minimizes the pressure of the softened PTC on the support PTC, hence reducing friction, further aided by the hydrogel lubrication layer. The continuum robot used in our study is 400 mm long, and throughout all experiments, we observed no issues with deployment due to frictional resistance. This length satisfies certain medical requirements. Third, due to the minimal flexural strength of the softened PTC, it buckles under resistance, ensuring that tissue damage caused by excessive pressure and popping forward is avoided. We have added the text below in the revised manuscript (**Line 377 to 392 Page 20-21**). “Although we initially assessed tissue damage qualitatively in the porcine esophagus, we further conducted in vitro experiments to quantitatively compare our robot's lateral force with that of a standard commercial catheter (ID=3.5mm, OD=4.9, Fresenius Kabi India Pvt. Ltd.) in a large angular position (**Fig. S12a, b, d, and Movie S14**). While our continuum robot can navigate a 180° bend without sidewall contact under precise positional control, we specifically tested the worst-case scenario where the robot remains close to the sidewall. The results show that the force exerted by our robot on the sidewall is nearly 100 times lower than that of the commercial catheter. Furthermore, we compared our robot's sidewall force with an existing LMPA-based continuum robot with variable stiffness (**Fig. S12b, c**), where both PTCs were simultaneously softened, advanced, and the sidewall forces recorded. Our robot exerted 10-15 times less force than the conventional variable stiffness robot. Additionally, the phase transition minimizes the pressure of the softened PTC on the support PTC, hence reducing friction, further aided by the hydrogel lubrication layer. The continuum robot used in our study is 400 mm long, and throughout all experiments, we observed no issues with deployment due to frictional resistance. This length satisfies certain medical requirements. Moreover, as the Guider becomes flexible before extending in each motion cycle, due to the minimal flexural strength, it buckles under resistance, ensuring that tissue damage caused by excessive pressure and popping forward is avoided.”

Figure S12. Force on the sidewall during a 180° turn. (a) Measuring the force on the sidewalls of a continuum robot as it moves forward. One side of the pressure sensor is fixed on the pedestal and the other side is connected to the corner sidewall. To avoid the effect of friction, the corner sidewall is not in contact with the pedestal. The force of the robot on the sidewall is captured by the sensor. (b) Pressure on the corner wall when two PTCs are alternately advanced. (c) Pressure on the corner wall when both PTCs are softened and advanced at the same time. (d) Pressure on corner wall when commercial catheter (OD=15Fr) is advanced. Scale bars, 10 mm.

■ **Comments #2:**

Related to the previous point, the parallelism between the continuum robot and the natural bine is not clear, including the comparison between the attraction towards light and an external magnetic field (Fig. 1a). In the former case, light serves as sensory feedback, while in the latter, a magnetic field is involved.

■ Response #2:

We apologize for the ambiguity caused by the expression. In **Fig. 1a**, we intended to exhibit the similarity of the motion of our continuum robot to that of a bine. Just as light influences the direction of a bine's growth, a magnetic field guides the movement of our robot. To provide readers with a more intuitive understanding of our robot's motion while ensuring textual accuracy, we have modified the term from 'growth-driven' to 'growth-inspired' and have scaled back on the elaboration of plant growth principles.

■ Comments #3:

The paper neglected a closely related technology, similarly based on two flexible devices supporting each other in an alternate fashion based on stiffness switching. Some related manuscripts are provided below as examples: Choset, Howie, and Wade Henning. "A follow-the-leader approach to serpentine robot motion planning." *Journal of Aerospace Engineering* 12.2 (1999): 65-73; Gilbert, Hunter B., Joseph Neimat, and Robert J. Webster. "Concentric tube robots as steerable needles: Achieving follow-the-leader deployment." *IEEE Transactions on Robotics* 31.2 (2015): 246-258; Kang, Byungjeon, Risto Kojcev, and Edoardo Sinibaldi. "The first interlaced continuum robot, devised to intrinsically follow the leader." *PloS one* 11.2 (2016): e0150278; Gao, Yuanqian, et al. "Continuum robot with follow-the-leader motion for endoscopic third ventriculostomy and tumor biopsy." *IEEE Transactions on Biomedical Engineering* 67.2 (2019): 379-390.

It is necessary to properly frame the proposed robot relative to close prior art: the introduction and discussion should be consistently revised.

■ Response #3:

We appreciate the reviewer for providing the relevant literature and carefully reading the content. In our research, we have thoroughly analyzed and compared existing robots with 'Follow the Leader' (FTL) capabilities, acknowledging the significant contributions made by researchers in this field. Works such as those by Choset et al., Gao et al., Gilbert, and Kang have laid important groundwork for developing FTL systems. The works of Choset et al., and Gao et al. involve linking several independent active deformation segments, providing more degrees of freedom (DOF) to enable FTL motion. However, these robots are typically larger and restricted to limited paths. Gilbert's concentric tube robots, although smaller in size, are limited to FTL movement along predefined paths. Kang's robot theoretically offers FTL capability over arbitrary trajectories but is hindered by its large size and lack of flexibility. In contrast, our robot demonstrates capabilities that differ significantly from these examples, as we have showcased in our paper, particularly in terms of size, flexibility, and the ability to navigate complex paths. We have now added the text below in **Line 43 to 56, Page 2-3** "Therefore, continuum robots should have the ability to actively comply with the features of the lumina and reach the target independently of tissue interaction. Through growth, bines can navigate complex terrains without relying on forces generated by interaction, which fulfills the above-mentioned requirement. Growing robots with similar growth principles to bines can retain the form of their previously grown segments while navigating (24,25), but suffer from large scale, low flexural strength, and long deployment times. Robots with the 'follow the leader' (FTL) behavior can also be deployed like bine growth. The FTL continuum robots are thus well-suited for navigation through the patient's anatomy while avoiding sensitive areas. FTL behavior is defined as the motion of a robot body along a path led by the tip (26) and can be implemented by concatenating several independent active deformation segments for more degrees of freedom (DOF). However, due to the difficulty of arranging deformation segments as compactly as muscles, these robots can only achieve FTL deployment along limited or predefined trajectories (27-29). Another FTL continuum robot is based on alternating deployment of two concentrically arranged snake robots with shape-lock mechanism (30), but the complex locking structure leads to large cross-section sizes and small angulation range."

■ Comments #4:

It is not fully clear how the robot was knotted: how was the magnetic field gradient produced? How large was such a gradient? It seems that quite a large gradient would be needed to achieve the knotting reported through main figures and SI. The ms should consistently reconsider whether the knotting capability could be practically implemented.

■ Response #4:

We apologize for not providing detailed descriptions of generating gradient forces. We use permanent magnets (50 x 50 x 50 mm) to generate gradient forces. The magnetic flux density gradient was a minimum of 1.5 T/m and a maximum of 23.1 T/m along the axis within 50 mm from the surface of the permanent magnet. Nan et al. used a tension force of 0.1 N to tie the knot (**K. Nan, et al., Nat. Biomed. Eng. 6, 1092-1104**). Through simulation, we determined the working distance of the permanent magnet. Moreover, the lubricating layer on the surface of our robot reduces the force required to tie the knot. Longer working distances can be achieved by using larger permanent magnets. We have made the following changes in the revised manuscript (**Line 269 to 273, Page 14**) “We used a 50 x 50 x 50 mm permanent magnet (N52) (**Fig. S5a**) to generate the gradient field. Magnetic flux density and magnetic flux density gradient around the permanent magnet were measured (**Fig. S5b**). The magnetic flux density gradient was a minimum of 1.5 T/m and a maximum of 23.1 T/m along the axis within 50 mm from the surface of the permanent magnet. We found through simulation that within 50mm, the force could be generated to tie a knot (**Fig. S5c**).”

Figure S5. Gradient force generated by the permanent magnet on the tip. (a) A magnetometer is mounted on a displacement stage to measure the magnetic field around a permanent magnet. (b) Magnetic flux density and magnetic flux density gradient versus distance from the surface of the permanent magnet (50×50×50 mm) at the neutral axis. (c) Simulation results of the pull force on the magnet at the tip of the robot as a function of distance.

■ Comments #5:

The actuation unit (i.e., that one pushing the PTCs) should be further detailed (and possibly shown).

■ Response #5:

We thank the reviewer for the suggestion to add a more detailed presentation of the advancement unit. We have now added Figure S20 to show the detailed advancement unit.

Figure S20. Structure and operational steps of the advancement unit. The advancement unit is composed of a linear module, a rotary motor, a transmission mechanism, a clamping mechanism, and a locking mechanism. The operational steps are as follows: i) The PTC is secured using the clamping mechanism, and then the linear module drives the PTC along the axial direction. ii) Upon reaching the limit position, the clamping mechanism is released and the locking mechanism is engaged. iii) With the PTC held in place by the locking mechanism, the linear module retracts to its initial position, ready for the next cycle of operation.

■ Comments #6:

The operating temperatures should be clearly reported. Indeed, considering the thermal strategy at the core of the proposed robot, it is essential to clearly state which is the temperature effectively reached by the PTCs during their operation. If the current working temperatures are relatively high for clinical applications (as the mentioned 47deg could be), the associated limitation should be clearly stated. (A complementary recording with a thermal camera might add value.)

■ Response #6:

In Figure S2d, we have presented the surface temperature measurements of the robot (42 °C), recorded in air at a temperature of 37°C. Notably, due to the good thermal conductivity of water, in an in vivo environment, the surface temperature of the robot is likely to be lower, influenced by the presence of body fluids. To provide a more comprehensive understanding, we also conducted thermal imaging of the robot's surface during operation in an ambient air temperature of 20°C. These recordings, captured with a thermal camera, show that the surface temperature of the continuum robot is approximately 40°C (**Fig. S4b**). This temperature range is within the biosafety standards for clinical applications (**Langberg, et al., Circulation 86, 1469-1474**). We have made the following changes in the revised manuscript (**Line 402 to 405, Page 21**). “Additionally, the PTC's thermal management capability ensures that the surface temperature remains below 42°C in air (refer to **Fig. S2d and Fig. S4b**), which is even lower in liquid environments, thus keeping within the biocompatible temperature range (35). In summary, these features enhance the clinical safety of our robot, making it a promising tool for medical applications.”

Figure S4. PTC with segmented variable stiffness capability. (a) Internal heating circuit of PTC. An additional resistive heater is added. (b) First, the resistance heater 2 was heated to soften segment 2. Then the PTC was bent and cooled under magnetic torque. Next, a suitable current was passed to the heating resistive heater 1 so that segment 1 was flexible while segment 2 remained rigid. Finally, a magnetic field was applied to bend the tip of the PTC. Scale bars, 10 mm.

■ **Comments #7:**

Some sentences are hard to understand (e.g., on line 166, “The advance distance should also not be less than 20 mm to balance the efficiency of the movement.”). Subplots e and f in fig 5 seem to be switched. In the title of Section 2.8, both “ex vivo” and “in vivo” should be mentioned. Some typos should be amended.

■ **Response #7:**

We apologize for the mistakes in language. We have addressed the reviewer's comments and made the following changes:

- 1). In Line 187, the sentence has been clarified to read: “To maintain movement efficiency, the advancing distance in each step should be no less than 20mm.”
- 2). Figure 5 has been corrected, ensuring that subplots e and f accurately reflect the described content.
- 3). The title of Section 2.8 has been revised for clarity to: “2.8 Clinical applications in ex vivo and in vivo studies assisted by X-ray imaging”
- 4). Additionally, we have scrutinized the manuscript and corrected several typographical errors.

Response to comments from Reviewer #3

I. General Comments:

In this manuscript, the authors introduce a novel continuum robot inspired by biological growth mechanisms, enabling apical extension while maintaining structural stability. Guided by programmable magnetic fields and incorporating phase transition components, the proposed robot can autonomously undergo cycles of tip-based elongation, featuring a solid-like backbone for stability and a liquid-like component for controlled advancement. The authors then demonstrate the robot's capability in navigating through complex luminal environments and thus implying its biomedical applications in in vivo imaging. The article is well written ensuring a fluent read, and the figures are well drawn. In light of this, the reviewer recommends the article be accepted for publication in Nature Communications before addressing the comments below:

■ Response:

We are grateful for the reviewer's supportive recommendation for publication and appreciate the positive feedback regarding our manuscript. We have diligently addressed the comments and concerns raised in the revised manuscript.

II. Major issues:

■ Comments #1:

Can the authors benchmark the specifications of the proposed robot, such as maximum length, diameter, angulation range, and motion speed, with state-of-the-art commercial endoscopes? A comparative analysis of these parameters with other reported technologies and existing commercial endoscopes should be provided. Any clear limitations of this system should be discussed with potential solutions.

■ Response #1:

We appreciate the reviewers' comments on the comparison with existing commercial endoscopes. In the revised manuscript, we conducted a comprehensive comparison of our robot with commercial endoscopes across various medical disciplines (**Table 1**) (**Line 75 to 80, Page 4**). Our robot matches commercial standards in key parameters such as diameter, length, and angulation range, and additionally offers growth-like safe motion and the ability to form functional structures. Due to the unavailability of specific motion speed data for commercial endoscopes, a direct comparison in this aspect was not feasible. One of the limitations identified in our robot is the prolonged deployment time. However, we are actively exploring solutions to reduce this, such as the implementation of active heating and cooling methods (**Line 451 to 454, Page 24**). “Although the advanced hybrid control strategy will reduce the surgery time and lower the working temperature below ambient temperature (46), adding the miniature cooling structure is still challenging. In future work, we will develop corresponding preparation methods.”

Table 1 Performance comparison of our robots with existing continuous robots

	Ref.	Actuation strategies	Diameter (mm)	Length (mm)	Angulation range (°)	Variable stiffness	Growth-like (FTL) deployment	Functionality	Application Scope	In vivo validation
Variable Stiffness Robot	Kim [10]	Magnetic	0.8	~200	±161	NO	NO	Carrying tools	Vascular intervention	Yes
	Lussi [18]	Magnetic	1	30	~ ±180	Thermal response	NO	Carrying tools	Ophthalmic Surgery	NO
	Mattmann [19]	Magnetic	1.3	140	-	Thermal response	NO	Carrying tools	Stomach and vessels	NO
	Chautems [20]	Magnetic	2.33	50	~ ±180	Thermal response	NO	Carrying tools	Vascular intervention	NO
	Piskarev [21]	Magnetic	2	88	±51	Thermal response	NO	Carrying tools	Vascular intervention	NO
FTL Robot	Amanov [31]	Tendon-driven	7	165	-	NO	Limited path	-	-	NO
	Gao [28]	Tendon-driven	3.4	60	~ ±180	NO	Limited path	-	Brain surgery	NO
	Gilbert [27]	Concentric tube	2.18	120	-	NO	Limited and predefined path	-	Brain surgery	NO
	Peyron [29]	Concentric tube+ Mag	1.07	213.5	±136.4	NO	Limited and predefined path	-	Nasal cavities and ear canals	NO
	Kang [30]	Multi-backbone	30	<200	±90	Mechanical locking	Arbitrary path	-	-	NO
This work	Magnetic	< 4	>400	~ ±180	Thermal response	Arbitrary path	Formation of functional structure or carrying tools	Vascular intervention and gastrointestinal surgery	Yes	
Commercial Catheters	Video cholangioscope (Innovex Medical Co.)		5	380	± 180	NO	NO	Carrying tools	Gastrointestinal tract	Yes
	Video bronchoscope BF-1TH190 (Olympus Co.)		6.2	600	± 120	NO	NO	Carrying tools	Bronchial tube	Yes
	Fiber pharyngoscope FS2 (Orlvision Co.)		2.9	300	± 130	NO	NO	Carrying tools	Nasopharynx	Yes
	Video cysto-nephroscope CYF-VH (Olympus Co.)		5.5	380	-130~220	NO	NO	Carrying tools	Urology	Yes

Compared with existing continuum robots, our robot, with small scale and flexibility, can achieve growth-like deployment along arbitrary paths. In addition to carrying tools, our robot can also form itself into functional structures in situ. The diameter, length, and angulation range of our robot have reached the level of commercial endoscopes. Through experiments, we found that our robot exerts 100 times and 10-15 times less force on the sidewall compared to commercial catheters and conventional variable stiffness robots, respectively.

■ **Comments #2:**

The reviewer finds the current Guide-Follower design quite cumbersome and space-occupying that may not be ideal for device fabrications as well as for operation under tight space such as in blood vessels. Can the authors discuss if there are any ways to integrate these two components and save space? For example can multilayer concentric tubes be used? Also, when integrating with fiber optics, can the fibers be inserted into the robot to further reduce the overall diameters?

■ **Response #2:**

We appreciate the reviewer's insightful comments. Adopting a concentric design could indeed reduce the overall size of the robot, though this approach significantly increases the complexity of fabrication. We are actively exploring this possibility in our ongoing research on concentrically arranged robots. While integrating fiber optics into the robot's design could potentially decrease its

diameter, this integration would likely result in an increase in the robot's bending strength, which is a critical factor for operation in constrained spaces. To balance size and functionality, we have opted to incorporate a miniaturized camera equipped with LED lighting in the current design.

■ Comments #3:

The authors should do a chronic leakage test under simulated physiological environments and provide quantitative data to show the successful sealing of cytotoxic components.

■ Response #3:

In response to concerns about potential leakage and cytotoxicity, we conducted a comprehensive three-day cytotoxicity test under simulated physiological conditions. The results indicated that our robot does not exhibit cytotoxic properties. Additionally, we evaluated the cytotoxicity of Bolton 117, the low melting point alloy (LMPA) used in our robot. Even in scenarios simulating a leakage, Bolton 117 demonstrated no significant cytotoxic effects. As a precautionary measure, we suggest the use of an additional sealing tube in practical applications to enhance safety. Furthermore, the integrity and state of the continuum robot can be effectively monitored by measuring its electrical resistance.

These findings and considerations have been detailed in the manuscript (**Line 393 to 405, Page 21**) “Our experiments provide substantial evidence of the biocompatibility of our robot. Initially, as demonstrated in **Fig. S13**, the PTC with its lubrication layer has been verified to be biocompatible. In the unlikely event of the lubrication layer being compromised, the PTC itself does not exhibit cytotoxic properties. Moreover, even in scenarios of potential leakage, the LMPA used in our robot does not show significant cytotoxicity. An additional layer of silicone tubing can be employed to ensure that any leaking LMPA remains isolated from the environment. Furthermore, any leakage would result in an increase in the electrical resistance of the PTC, as the cross-sectional area of the LMPA decreases. This change in resistance can be monitored to assess the state of the continuum robot (as shown in **Fig. S14**). It is also noteworthy that the design of our robot minimizes the presence of sharp edges that could potentially damage luminal structures. Additionally, the PTC's thermal management capability ensures that the surface temperature remains below 42°C in air (refer to **Fig. S2d and Fig. S4b**), which is even lower in liquid environments, thus keeping within the biocompatible temperature range (35). In summary, these features enhance the clinical safety of our robot, making it a promising tool for medical applications.”

The experimental details of the cytotoxicity test are as follows (**Line 595 to 602, Page 30**) “To evaluate the cytotoxicity of our continuum robot, live/dead cell staining was used to detect cell viability. The human umbilical vein endothelial HUVEC cells were cultured in Dulbecco’s modified Eagle’s medium (DMEM) (Gibco) supplemented with 10% (v/v) fetal bovine serum (Biological Industries). PTC with hydrogel layers, PTC without layers, and LMPA as cytotoxicity assay objects were used as cytotoxicity assay objects and placed in the culture environment of HUVEC cells, respectively. Cells cultured in medium only were used as positive controls. All cells were grown at 37°C in the humidified incubator with 5% CO₂. Cells around the test object were taken every 24 hours and stained with Calcein AM. The cells were washed once with PBS and observed under the fluorescence microscope by the excitation wavelength of 488nm.”

Figure S13. Cytotoxicity Test of PTC and LMPA on HUVEC Cells. PTC with hydrogel layers, PTC without layers, and LMPA were placed in the culture environment of HUVEC cells for 3 days for cytotoxicity test. Scale bars, 100 μm .

Figure S14. Leakage detection in PTC. If the PTC is punctured, the pressure inside it will squeeze out the LMPA, causing the radial dimension to decrease and the resistance to rise. If an abnormal rise in resistance is detected, the continuum robot should be withdrawn quickly.

■ **Comments #4:**

What is the size limitation of this approach i.e. how small can it go? What are the associated challenges and potential solutions

■ **Response #4:**

We appreciate the important and valuable comment from the reviewer. In our discussion in Fig. 2c, we have addressed the size relationship between the Follower and Guider necessary to accomplish growth-like motion. If the effect of gravity on shape is not a concern, the radial dimensions of the PTCs can be further reduced for more minuscule and tortuous body lumina, such as cerebral vasculature. However, Due to the thickness of the silicone tubing encapsulating the LMPA, the ratio of flexural strength of the PTC in the rigid and flexible state reduces as the diameter decreases. To ensure the interlocking effect of the Follower and Guider, as well as the space required for the heating circuit, the diameter of the PTC should be larger than 0.5mm. The overall size of the robot thus

can be as small as about 1mm. Further size reductions can be achieved by using encapsulated tubes with thinner wall thicknesses or by using non-encapsulated variable stiffness materials with large changes in modulus of elasticity. We have added the discussion of the minimum size of the PTC in **Line 492 to 495, Page 25-26** “Due to the thickness of the silicone tubing encapsulating the LMPA, the ratio of flexural strength of the PTC in the rigid and flexible state reduces as the diameter decreases. To ensure the interlocking effect of the Follower and Guider, as well as the space required for the heating circuit, the diameter of the PTC should be larger than 0.5mm.”

■ **Comments #5:**

Although the robot does not require touching the lumina to move, it seems from Figure 6-8 that during actual in vivo operations the device inevitably contacts the tissues, which may eliminate the claimed advantages such as reduced potential medical risks and restricted mobility. The authors should discuss strategies to ensure little or no contact of the robot with the lumina throughout the in vivo operations.

■ **Response #5:**

We thank the reviewer for the comment. While it is true that our robot may occasionally come into contact with surrounding environments during deployment due to control precision or environmental factors, it's important to note that these interactions are not relied upon for the robot's deformation. We conducted tests to measure the force exerted by our robot on the sidewall during such contact, and the results confirm that the force exerted by our robot on the sidewall is nearly 100 times lower than that of the commercial catheter and 10-15 times lower than that of the conventional variable stiffness robot (**Fig. S12**). Therefore, the contact of our robot with its environment does not constitute a safety risk. Enhancing the accuracy of positioning and reducing the length of the protruding segment in each cycle can further improve the control accuracy of the magnetic field, thereby minimizing the robot's contact with its surroundings. Additionally, dynamic adjustment of control parameters during deployment helps the robot adapt to complex environments and avoid unnecessary contact.

Figure S12. Force on the sidewall during a 180° turn. (a) Measuring the force on the sidewalls of a continuum robot as it moves forward. One side of the pressure sensor is fixed on the pedestal and the other side is connected to the corner sidewall. To avoid the effect of friction, the corner sidewall is not in contact with the pedestal. The force of the robot on the sidewall is captured by the sensor. (b) Pressure on the corner wall when two PTCs are alternately advanced. (c) Pressure on the corner wall when both PTCs are softened and advanced at the same time. (d) Pressure on corner wall when commercial catheter (OD=15Fr) is advanced. Scale bars, 10 mm.

REVIEWER COMMENTS

Reviewer #2 (Remarks to the Author):

The authors have significantly enhanced the clarity and quality of the manuscript based on the suggested remarks. However, there are still some aspects that require revision in the current version of their work.

While the authors have minimized the emphasis on bioinspiration related to plant growth in explaining the proposed follow-the-leader approach, references to growth persist at various points in the manuscript, including the title. It is noteworthy that the follow-the-leader approach has been proposed for over forty years, and the concept of bioinspiration from plant growth has never been mentioned before.

Additionally, the response to my previous comment regarding the parallelism between the continuum robot and the natural bine is not yet thoroughly clear. As previously indicated, the comparison between the attraction towards light and an external magnetic field (Fig. 1a) is inaccurate, given that light serves as sensory feedback, whereas in the latter, a magnetic field is involved. Therefore, the term 'growth-inspired' is not an appropriate definition.

These points need careful modification before submission.

Reviewer #3 (Remarks to the Author):

The authors have satisfactorily addressed all comments and this reviewer recommends the manuscript to be accepted for publication in Nature Communications.

Responses to Comments on “NCOMMS-23-43861A-Z”

We sincerely thank two reviewers for the valuable comments and feedback provided during the review process. Your suggestions have been carefully considered and incorporated to further refine and improve our manuscript. For the convenience of the reviewers, the corresponding modifications are highlighted in purple font in this letter and with yellow background in the revised version.

Response to comments from Reviewer #2

■ Comments:

The authors have significantly enhanced the clarity and quality of the manuscript based on the suggested remarks. However, there are still some aspects that require revision in the current version of their work.

While the authors have minimized the emphasis on bioinspiration related to plant growth in explaining the proposed follow-the-leader approach, references to growth persist at various points in the manuscript, including the title. It is noteworthy that the follow-the-leader approach has been proposed for over forty years, and the concept of bioinspiration from plant growth has never been mentioned before.

Additionally, the response to my previous comment regarding the parallelism between the continuum robot and the natural bine is not yet thoroughly clear. As previously indicated, the comparison between the attraction towards light and an external magnetic field (Fig. 1a) is inaccurate, given that light serves as sensory feedback, whereas in the latter, a magnetic field is involved. Therefore, the term 'growth-inspired' is not an appropriate definition.

■ Response:

Your recognition of the significant improvements we have implemented is greatly appreciated and has encouraged us to further refine our work with even greater diligence. In light of your latest feedback, we have taken a thorough approach to addressing the remaining aspects that require revision. We acknowledge your concerns about the use of "growth-inspired" terminology and the comparison between the natural growth of bine and our continuum robot. We are committed to ensuring that our manuscript accurately reflects our research while adhering to the highest standards of scientific communication. To this end, we have made a series of targeted adjustments in response to your specific points.

The primary goal of our work is to address the limitations of traditional continuum robots used in transluminal procedures, which achieve passive bending through interaction with luminal walls, potentially causing tissue damage and limiting movement performance. Inspired by the observation that bines grow and navigate in complex environments without relying on external forces, we aim to develop a continuum robot with similar movement capabilities. To achieve this, our design incorporates two phase-changing components with interlocking shapes, enabling apical extension and secure structural stability. This movement, reminiscent of bine growth in time-lapse photography, initially led us to describe our robot using terms such as "growth-inspired" or "growth-like." However, recognizing that our robot falls within the follow-the-leader (FTL) category, we have revised our manuscript to remove references to growth-related concepts for accuracy within the appropriate scientific context. Consequently, we revise the title to “**Magnetic Steering Continuum Robot for Transluminal Procedures with Programmable Shape and Functionalities**”. We have also updated **Figure 1** to de-emphasize the growth-inspired concept, focusing instead on directly explaining its motion mechanism. We now only mention the similarity to bine movement when describing the robot's movement effect for intuitive understanding.

It is important to emphasize that our continuum robot, compared to existing robots based on various FTL methods, is smaller in size and capable of perfect FTL deployment along any chosen path. These significant advantages expand the potential applications of such robots, including in-situ formation of controllable shapes as functional tools, leveraging shape programmability.

We have made the following revisions to the manuscript and supplementary material.

Line 4 to 6, Page 1 “Here, we introduce a millimeter-scale continuum robot, enabling apical extension while maintaining structural stability.”

Line 43 to 53, Page 2 and 3 “Therefore, continuum robots should have the ability to actively comply with the features of the lumina and reach the target independently of tissue interaction. Robots exhibiting the 'follow-the-leader' (FTL) behavior offer a promising solution, as they can operate without relying on environmental interactions. This behavior enables FTL continuum robots to navigate through the patient's anatomy adeptly while avoiding delicate regions. FTL behavior, as herein defined, denotes the movement of a robot body along a trajectory guided by its tip (24), which can be realized by integrating multiple independent active deformation segments to enhance degrees of freedom (DOF). However, the challenge lies in arranging these deformation segments as efficiently as biological muscles, consequently limiting FTL deployment to specific or pre-defined trajectories (25-27). An alternative approach to FTL continuum robotics involves the deployment of two concentrically arranged snake robots with a shape-lock mechanism (28). Nonetheless, the intricate locking structure results in increased cross-sectional dimensions and restricted angulation range.”

Line 54 to 62, Page 3 “Here, we put forth a millimeter-scale continuum robot with FTL behavior, capable of apical extension and secure structural stability (**Fig. 1a** and **b**). Utilizing a dual-component system based on phase transitions, the robot engages in periodic, tip-based elongation steered by a programmable magnetic field. Each of these motion cycles integrates a stable, solid-like backbone along with a liquid-like component for forward advancement, allowing the robot's shape to be actively programmed or reprogrammed through trajectory planning of its tip, independent of environmental interactions. When integrated with advanced imaging technologies, our robot is capable of achieving precise magnetically guided navigation, akin to a bine threading through narrow and intricate lumina, while significantly reducing tissue damage and friction (**Fig. 1c i**).”

Line 80 to 83, Page 4 “The detailed structure of our continuum robot is depicted in Figure 1a. We achieved the FTL behavior of the proposed continuum robot by alternately advancing a couple of phase transition components (PTCs) that are constrained to move only axially relative to each other (**Fig. 1b**). These PTCs can transition between solid and liquid states, altering their stiffness to switch roles between support and movement.”

Line 96 to 107, Page 5 “The Guider and Follower operate in an alternating manner, forming a cooperative relationship as follows: (1) Initially, the Guider, heated to the phase-transition temperature, becomes flexible for forward movement, while the Follower remains rigid to provide support. With multiple orders of magnitude differences between their flexural strengths (Fig. 2c, detailed structure analysis provided in Materials and Methods), the rigid Follower maintains the shape of the deployed segment of the continuum robot, serving as a conduit to deliver the flexible Guider forward under proximal thrust. The flexible outstretched segment of the Guider will be deflected toward the direction of the magnetic field. (2) Subsequently, the Guider and Follower switch states, transitioning their roles within the motion cycle of the robot. The now rigid Guider stabilizes the deployed shape, while its protruding segment forms a new conduit, constraining the shape of the propelled flexible Follower. The maximum cross-sectional size of the robot is approximately 3-4 mm, with a potential length extending up to meters. For demonstration purposes, we employ a robot with a length of approximately 400 mm (Fig. S1d).”

Line 188 to 198, Page 10 “In the motion cycle, the Guider first undergoes a controlled heating process to achieve flexibility under PID control. Once its total resistance change stabilizes, indicating the attainment of the flexible state, the Guider is propelled forward by the advancement unit and steered in response to the applied magnetic field. Subsequently, both the advancement unit and the

heating circuit are deactivated, while the magnetic field is maintained until the Guider transitions from flexibility to rigidity. Following this, the Follower is heated to flexibility and advanced along the trajectory of the Guider by the advancement unit. Acknowledging potential errors in the kinematic model and practical contingencies, priority is accorded to operator inputs to the experimental equipment during pre-programmed movements of the robot. Ultimately, our robot demonstrates the capability of follow-the-leader (FTL) deployment along arbitrary paths, exhibiting accessibility and dexterity reminiscent of a bionic arm. It can even navigate complex terrains, including climbing and curling around branches (Fig. 3e and Movie S3).”

Line 214 to 215, Page 12 “Under magnetic steering, the proposed continuum robot, capable of FTL behavior, can navigate along planned paths through highly unstructured environments with little or no interactions (Fig. 4a).”

Line 240 to 246, Page 13 “Our focus extends beyond the inherent challenges and requirements faced by traditional soft continuum robots. We also explore the unique capability of our robot to actively and dynamically program and reprogram its entire body in situ, transforming into a functional instrument. Upon reaching the target site safely, such as the bladder, ventricles, or abdominal cavity, the robot can continue to progress along a predetermined path, morphing into complex and functional structures suitable for various surgical tasks or sensing applications. This capability allows the robot to overcome the limitations, that the small inner diameter of natural lumina or access ports imposed, on the geometries and functionalities of surgical tools (Fig. 5a).”

Line 286 to 289, Page 16 “To validate the potential for clinical translation, we combined the robot with common medical technologies to demonstrate the navigation of our robot in biologically relevant phantoms, organs ex vivo, and environment in vivo. First, we extended the demonstrated magnetic navigation in FTL manner to a more realistic, clinically relevant environment in vitro, assisted by ultrasound imaging.”

Line 411 to 420, Page 23 “The imperative need for small-scale soft continuum robots to autonomously extend apically while maintaining structural stability, without relying on forces generated by environmental contact, is paramount for enhancing surgical safety, reducing complexity, and expanding functional capabilities. In this study, we introduce a magnetic steering continuum robot exhibiting follow-the-leader (FTL) behavior to address this critical requirement. Our robot consists of two PTCs that can only move axially in relation to each other, alternately changing their stiffness and moving in periodic. Throughout each motion cycle, these components alternate between serving as a stable, solid-like backbone and a liquid-like element for progressive advancement. This design enables the robot's shape to be actively and dynamically programmed or reprogrammed through trajectory planning of its tip, thereby enabling operation independent of environmental interactions.”

Figure 1. Schematic of the millimeter-scale magnetic steering continuum robot for transluminal procedures. (a) Overview of the proposed continuum robot. Our robot consists of a couple of phase transition components (PTCs) named “Guider” and “Follower”. The Guider is assembled coaxially in a silicone tube bonded to the Follower, and its tip has an embedded tiny permanent magnet to respond to external magnetic fields. The low melting point alloy (LMPA) and the resistive heater are encapsulated by the silicone tube, and the hydrogel coating is grown on the surfaces of the Guider and Follower to reduce the friction coefficient significantly. **(b)** Schematic of our robot’s design principle. The shapes of the Guider and Follower are interlocked. In the motion cycle, the two PTCs alternate between solid and liquid, corresponding to the states of support and movement. **(c)** Under magnetic steering, the magnetic steering continuum robot shows excellent movement performance **(i, ii, iii)** and functionalities **(iv, v, vi)** in lumina or the relatively open cavities of organs.

Response to comments from Reviewer #3

■ **Comments:**

The authors have satisfactorily addressed all comments and this reviewer recommends the manuscript to be accepted for publication in Nature Communications.

■ **Response:**

We are grateful for your positive evaluation and recommendation. Your support is immensely important to us, and we are pleased to see that the revised manuscript has met your approval.

REVIEWERS' COMMENTS

Reviewer #2 (Remarks to the Author):

The authors have responded clearly and comprehensively to the reviewers' remarks and feedback, significantly improving the manuscript's quality.

Responses to Comments on “NCOMMS-23-43861B”

Response to comments from Reviewer #2

■ Comments:

The authors have responded clearly and comprehensively to the reviewers' remarks and feedback, significantly improving the manuscript's quality.

■ Response:

We sincerely thank the reviewer for the kind comments and helping us improve the quality.